# An Investigation of LLMs' Inefficacy in Understanding Converse Relations

**Chengwen Qi**[1,♣]    **Bowen Li**[2,3,♣]    **Binyuan Hui**[3]    **Bailin Wang**[3,5]    **Jinyang Li**[4]
**Jinwang Wu**[1]    **Yuanjun Laili**[1,†]

[1]Beihang University  [2]Shanghai AI Laboratory
[3]3B Group    [4]The University of Hong Kong    [5]MIT
chengwen_qi@buaa.edu.cn, libowen.ne@gmail.com
huybery@gmail.com, lailiyuanjun@buaa.edu.cn
https://github.com/3B-Group/ConvRe

## Abstract

Large Language Models (LLMs) have achieved remarkable success in many formal language oriented tasks, such as structural data-to-text and semantic parsing. However current benchmarks mostly follow the data distribution of the pre-training data of LLMs. Therefore, a natural question rises that do LLMs really understand the structured semantics of formal languages. In this paper, we investigate this problem on a special case, converse binary relation. We introduce a new benchmark ConvRe focusing on converse relations, which contains 17 relations and 1240 triples extracted from popular knowledge graph completion datasets. Our ConvRe features two tasks, Re2Text and Text2Re, which are formulated as multi-choice question answering to evaluate LLMs' ability to determine the matching between relations and associated text. For the evaluation protocol, apart from different prompting methods, we further introduce variants to the test text and few-shot example text. We conduct experiments on three popular LLM families and have observed various scaling trends. The results suggest that LLMs often resort to shortcut learning and still face challenges on our proposed benchmark.

## 1 Introduction

Large Language Models (LLMs) have demonstrated impressive empirical results on various NLP tasks (Bubeck et al., 2023; OpenAI, 2023; Anthropic, 2023), including formal language-oriented tasks such as structural data-to-text (Xiang et al., 2022) and semantic parsing (Chen et al., 2021; Li et al., 2023a), which require sophisticated comprehension and production of structured language content. Despite these promising advances, a critical concern remains largely unexplored: *do these LLMs genuinely understand the nuanced semantics*

*of formal languages, **or are they merely exploiting statistical patterns inherent in their pre-training data?*** If such shortcuts exist, it implies that LLMs may struggle to generalize to novel and unique formal language definitions, potentially hindering the robustness and scalability of practical applications.

In this work, we delve into this question by focusing on a specific aspect of formal language understanding: the comprehension of ***converse relations***. As shown in Figure 1, the converse relation redefines the semantic relation between entities while keeping the surface form of the triple unchanged. For instance, the triple $(x, \text{has part}, y)$ should be interpreted as *"x has a part called y"* in the normal relation (Codd, 1983), while *"y has a part called x"* in converse form. Notably, LLMs are largely unfamiliar with converse relations, as the data they learn in pre-training mostly comprises normal relation. It's imperative for LLMs to accurately understand and utilize these converse relations, i.e., truly following instructions rather than recalling memorized patterns (shortcuts) about normal relation, as it significantly impacts the semantic coherence of their output.

To systematically evaluate the competence of LLMs in recognizing and processing converse relations, we introduce a novel benchmark, **ConvRe**. This benchmark draws upon 17 diverse relations and 1240 triples derived from prominent knowledge graph completion datasets. ConvRe introduces two primary tasks, **Re2Text** and **Text2Re**, formatted as multiple-choice question answering tests. These tasks challenge LLMs to correctly match relations (**Re**) with their corresponding natural language text (**Text**).

During empirical evaluation, we add various prompting methods and introduce variants to the text. More specifically, we manually craft examples of different types in the few-shot prompting, creating a more challenging testbed for these models. Our findings, based on thorough experiments

---

♣ Equal contribution.
† Corresponding authors.

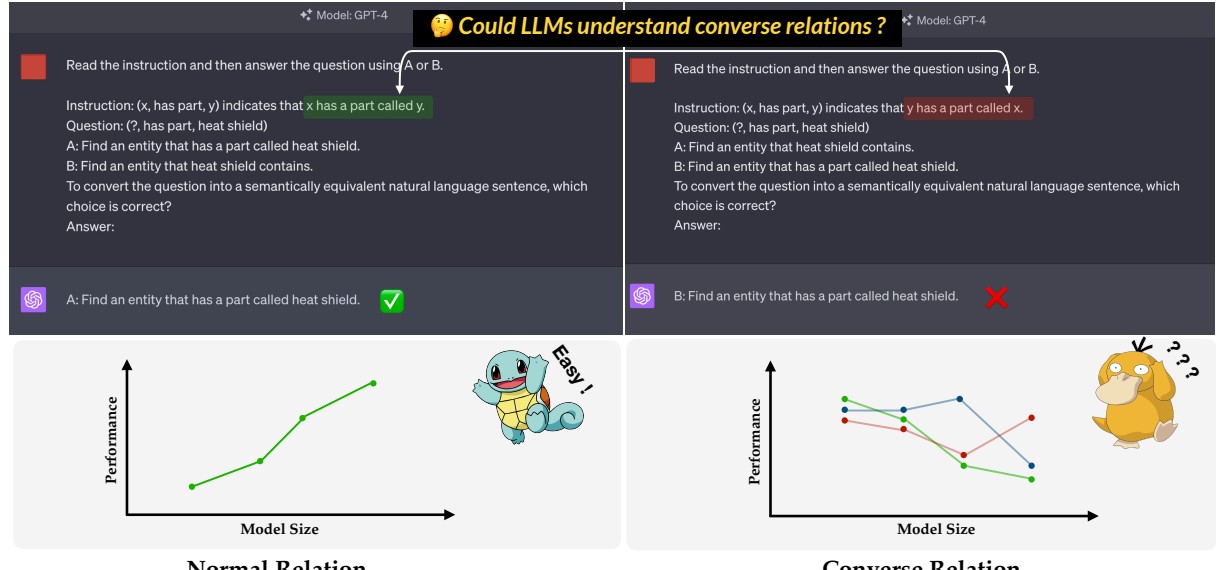

Figure 1: Illustration of converse relation comprehension by LLMs. This diagram highlights the unique challenges converse relations present for LLMs, potentially leading to diverse scaling trends.

| Triple | Relation | Notation | Associated Text | Text Variants |
|---|---|---|---|---|
| $(x, R, y)$ | Normal | $\mathbb{R}$ | $s$ 
 $x$ has a part called $y$. | $s'$ 
 $x$ possesses a specific component named $y$. |
| $(x, \mathtt{has\ part}, y)$ | Converse | $\mathbb{R}^\top$ | $s^\top$ 
 $y$ has a part called $x$. | $s^{\top\prime}$ 
 $y$ contains $x$. |

Table 1: The definition of normal and converse relation. Examples are provided below the notations. A triple can be defined to represent the normal relation $\mathbb{R}$ or the converse relation $\mathbb{R}^\top$. Each relation is associated with a pairing natural language text, which can further be paraphased.

using three popular LLM families, reveal interesting scaling trends and suggest that performance on understanding formal languages might be inflated by shortcut learning. This exploration contributes to the growing body of literature that seeks to assess the true capabilities of LLMs, and the extent to which they genuinely comprehend the semantics of formal languages.

## 2 ConvRe Benchmark

In this section, we will introduce the motivation, task formulation and design choice of our ConvRe benchmark as well as the details surrounding data collection.

### 2.1 Motivation

The recent surge in the performance of LLMs in understanding formal language, including tasks such as semantic parsing or data2text, can potentially be misleading. Traditional evaluation benchmarks used in such tasks often reflect statistical patterns similar to those found in the pre-training

data of LLMs. We posit that this could lead LLMs to take a *shortcut* [*] as described in Geirhos et al. (2020), thereby inflating the understanding of formal language semantics. Instead of comprehensively grasping the semantics, the LLMs might simply be learning the statistical tendencies present in their training data. To this end, we propose a new benchmark that uses normal and converse relations to examine the true semantic comprehension capabilities of LLMs.

### 2.2 Normal and Converse Relation

**Normal Relation**    Formally, a binary relation $\mathbb{R}$ over sets $X$ and $Y$ is a set of *ordered pairs* $(x, y)$ consisting of elements $x \in X$ and $y \in Y$ (Codd, 1983). Usually, a *normal relation* $\mathbb{R}$ is represented as $\mathbb{R} = \{(x, R, y) \implies xRy\}$, where $R$ is the specific relation phrase. Normal relations usually appear in the knowledge graph, along with a pair of

---

[*]There are some terminologies, such as *spurious correlation* and *superficial cues/bias/artifacts*, that are similar to the term *shortcut* used in this paper. We provide supplementary explanations of these terms in Appendix A for better clarity.

**Re2Text Task**

Read the instruction and then answer the question using A or B.

Instruction: (x, has part, y) indicates that y has a part called x.
Question: (?, has part, hilt)

LLM Pre-Training Corpus
**Triple**: (sword, has part, hilt)
**Text**: sword has a part called hilt

*shortcut*

A: Find an entity that has a part called hilt.
B: Find an entity that is a part of hilt.

*shortcut*

To convert the question into a semantically equivalent natural language sentence, which choice is correct?
Answer:
Expected answer: B

**Re2Text Task (hard)⭐**

Read the instruction and then answer the question using A or B.

Instruction: (x, has part, y) indicates that y has a part called x.
Question: (?, has part, hilt)

LLM Pre-Training Corpus
**Triple**: (sword, has part, hilt)
**Text**: sword has a part called hilt

*shortcut*

A: Find an entity that has a part called hilt.
B: Find an entity that hilt contains. (altered)

✗ *no shortcut*

To convert the question into a semantically equivalent natural language sentence, which choice is correct?
Answer:
Expected answer: B

Figure 2: The Re2Text task converts relation into semantically equivalent natural language text. Given that LLMs mostly encounter normal relations during pre-training, deciphering converse relations poses a significant challenge. LLMs tend to exploit textual similarity shortcuts for prediction, which can mislead the model's performance as it bypasses genuine comprehension. In the regular scenario (top), two shortcuts lead the model towards divergent answers, where the incorrect answer (A) will not be overly preferred. In the hard ⭐ scenario (bottom), the text for the correct response (B) is modified, transforming two shortcuts into a single one. This solitary shortcut is more likely to misdirect the model towards the incorrect answer (A), highlighting the pitfalls of shortcuts learning.

subject $x$ and object $y$. This triple can be mapped to a semantically equivalent natural language text $s$. Examples can be found in Table 1.

**Converse Relation**  In addition to the *normal relation*, we also introduce a *converse relation* that utilizes the same triple format $(x, R, y)$ to denote the converse mapping $\mathbb{R}^\top$. It defines a new form by swapping the pairing order, which can be expressed as $\mathbb{R}^\top = \{(x, R, y) \implies yRx\}$. Accordingly, in the converse mapping, the triple $(x, R, y)$ corresponds to the converse natural language text $s^\top$. Examples are provided in Table 1 for further clarity.

It's worth noting that both the normal and converse relation definitions used in our evaluation have a *localized scope* to minimize ambiguity. This process helps us ascertain whether LLMs can understand the semantics of the custom relation definition rather than resorting to shortcut learning.

### 2.3 Task Formulation

We designed two tasks to assess LLM's understanding of normal and converse relations. Both tasks focus on semantic equivalence translation between relations (**Re**) and natural language text (**Text**).

**Re2Text**  In this task, given the specification of a normal/converse relation and its associated natural language text along with a query triple, the model is asked to determine the natural language text that best aligns semantically with the query triple.

**Text2Re**  The second task can be considered as the reverse of Re2Text. Given an instruction—formatted similarly to Re2Text—and a query sentence, the model is required to identify the query triple that best matches the query sentence.

Following McKenzie et al. (2023), both tasks are formulated as multi-choice question-answering tasks, providing a concrete method for evaluation.

### 2.4 Text Variants

Geirhos et al. (2020) highlighted a phenomenon in deep learning known as *shortcut learning*. These are decision rules that achieve high performance on standard benchmarks but fail to generalize under more challenging testing conditions such as real-world scenarios. This issue is particularly significant in language processing tasks, where a language model may show an ability to reason that is learned from the training data, but its performance can drop drastically—sometimes to levels

Figure 3: The Text2Re task converts natural language text into semantically equivalent relation triple. As with the Re2Text task, this process can be misled by shortcut learning. In the regular scenario (top), an altered question is used, resulting in a single shortcut that leads the model towards the incorrect answer (A). In the hard ⭐ scenario (bottom), the combination of natural language text and the relation definition creates two shortcuts, both leading to the incorrect answer (A), thus increasing the likelihood of the model's misprediction.

equivalent to random guessing—when superficial correlations are removed from the dataset (Niven and Kao, 2019).

To assess how extensively current LLMs leverage shortcut learning for the evaluation tasks we have designed, we introduce variants to the text in both our tasks. Concretely, we alter the natural language text on both *test* side and *few-shot example* side to get the paraphrased variants.

**Test Variants** In the Re2Text task, we paraphrase one answer candidate, while in the Text2Re task, we paraphrase the question. Specifically, we modify the key predicate and restructure the sentence. We note that the subtle variations on the test text could bring different effects to the two tasks, which will be evidenced by the empirical results in our experiments (see Section 4.2). Examples on the test variants as well as intuitive explanations on their effects on two tasks are provided in Figure 2 and 3. Detailed zero-shot prompting methods can be found in Table 2.[†]

**Few-shot Example Variants** Beside the variants on the test text, we further introduce variants to the text within the examples used for few-shot prompting. Since we have identified the most challenging

variant settings within the zero-shot tasks, we will employ the same configurations for the test text in the few-shot context, denoting these as *hard* tests. Accordingly, we integrate text variants within the examples for the few-shot prompting. A comprehensive list of few-shot prompts utilized in our benchmark can be found in Table 3, and the specific arrangements of text variants are illustrated in Table 4. Notably, if the *hard* test setting aligns with the unaltered test text (for the Text2Re task), then the unaltered examples are labeled as *hard*, while the altered examples are labeled as *regular*. This setup shares the similar spirit as the complexity based prompting (Fu et al., 2022), where *hard* examples serve to refine problem understanding and mitigate model bias.

### 2.5 Data Collection

To make our tasks more comprehensive, and thus test the LLMs' ability to reason in more complex ways, plausible relations must satisfy two requirements:

- The relation is ***asymmetric***, implying that $\mathbb{R} \neq \mathbb{R}^{\top}$. An example of such a relation is parent of. Here, the order of the involved entities significantly changes the meaning, as the parent-child relationship is not mutual. Con-

---

[†]Relation settings and hint will be thoroughly discussed in Section 3.2.

| ID* | Prompting Method | Shot | Relation† | Hint | Test Variants‡ |
|---|---|---|---|---|---|
| 1# | normal-re, normal-text | 0 | N | | |
| 2# | normal-re, altered-text | 0 | N | | ✓ |
| 3# | converse-re, normal-text (⭐ Text2Re ) | 0 | C | | |
| 4# | converse-re, altered-text (⭐ Re2Text) | 0 | C | | ✓ |
| 5# | converse-re, normal-text, hint | 0 | C | ✓ | |
| 6# | converse-re, altered-text, hint | 0 | C | ✓ | ✓ |

Table 2: Zero-shot prompts. *: each prompt method has been associated with a unique ID that will be referred to in the experimental results. †: N indicates *normal* relation and C indicates *converse* relation. ‡: whether test text are altered. ⭐ Text2Re: the *hard* setting for Text2Re. ⭐ Re2Text: the *hard* setting for Re2Text.

| ID | Prompting Method | Shot | Relation | Hint | Examples♣ | Tests♠ |
|---|---|---|---|---|---|---|
| 7# | 3-shot, hard-hard | 3 | C | | hard | hard |
| 8# | 3-shot, hard-hard, hint-cot | 3 | C | ✓ (w/ CoT) | hard | hard |
| 9# | 6-shot, hard-hard | 6 | C | | hard | hard |
| 10# | 3-shot, regular-hard | 3 | C | | regular | hard |
| 11# | 3-shot, regular-hard, hint-cot | 3 | C | ✓ (w/ CoT) | regular | hard |
| 12# | 6-shot, regular-hard | 6 | C | | regular | hard |

Table 3: Few-shot prompts. ♠: the hard test setting is always employed (see Table 2). ♣: examples are provided in two options, regular and hard.

| Example-Test | Example Variants | Test Variants |
|---|---|---|
| *Re2Text* | | |
| hard-hard | ✓ | ✓ |
| regular-hard | | ✓ |
| *Text2Re* | | |
| hard-hard | | |
| regular-hard | ✓ | |

Table 4: Text variants on test and example sides for few-shot prompting.

versely, if the relation is symmetric, such as `neighboring country`, it would be meaningless to determine whether a given entity should be a head or a tail, as the both are semantically equivalent.

- The involved subject and object are ***interchangeable***. That is, the relation $\mathbb{R}$ and its converse counterpart $\mathbb{R}^\top$ should be semantically plausible, though not equivalent. An example of a relation we would avoid under this criterion is `native language`, which associates a person with a language. A language cannot logically be the subject of `native language`, thereby disqualifying this relation. Relations of this sort could allow LLMs to rely on shortcut learning to solve tasks. For instance, in the case of `native language`,

the entity's type inadvertently reveals the answer so that the LLMs may exploit this leaked information.

We manually select 17 relations from five widely used knowledge graph datasets: WN18RR (Dettmers et al., 2018), FB15K-237 (Toutanova and Chen, 2015), Wikidata5M (only transductive settings) (Wang et al., 2021), NELL-ONE (Xiong et al., 2018), ICEWS14 (García-Durán et al., 2018), ConceptNet5 (Speer et al., 2017). For each relation, we randomly sample 80 triples from corresponding datasets and manually remove the triples that are not suitable for our task. Finally, we get 1240 triples in our benchmark, detailed breakdown of the number of triples for each relation can be found in Appendix B.

## 3 Experiment Setup

### 3.1 Model and Metric

We evaluated three LLM families on our ConvRe benchmark: OpenAI GPT-3 (Brown et al., 2020), Anthropic Claude (Anthropic, 2023), and Google Flan-T5 (Chung et al., 2022) (model details in Appendix C). Since we do not have enough credits for the OpenAI APIs, we evaluate OpenAI GPT-4 on a subset of our benchmark for few-shot experiments.‡ We use the classification accuracy as our

---

‡The subset is constructed by randomly sampling 20 triples for each relation from the full set. In the case where the

| Prompt |
| --- |
| Read the instruction and then answer the question using A or B. Note that in this task, if the relation is defined in a converse manner, unlike the conventional definition, you should carefully choose the answer. |
| Instruction: (x, has part, y) indicates that x has a part called y.
Question: (?, has part, solingen)
A: Find an entity that solingen contains.
B: Find an entity that has a part called solingen.
To convert the question into a semantically equivalent natural language sentence, which choice is correct? Look out for the ORDER of the entities in the instruction!
Answer:
Expected Answer: B |

Figure 4: An illustration of zero-shot prompting with hint. Red color font indicates the hint.

main metric for both Re2Text and Text2Re tasks.

## 3.2 Prompting Methods

As depicted in Zhang et al. (2023), different prompting methods can have a considerable impact on the scaling trends of language models. To account for this in our study, we utilize diverse prompting methods. Generally, we have zero-shot and few-shot prompting, each tailored with specific design elements. Detailed illustrations are provided in Table 2, 3 and 4. While we previously discussed these from a motivation point of view, this subsection offers a closer look at the implementation specifics.

**Zero-shot** We assess both normal and converse relations mainly on the zero-shot setting, where each setting is coupled with regular and altered test text (refer to the text variations in Section 2.4). For the converse relation evaluation, we additionally equip the prompt with hint (Kojima et al., 2022). An illustration of the hint used in our experiment is shown in Figure 4.

**Few-shot** In this setting, we only apply the *hard* settings, as documented in Table 3. The corresponding zero-shot tests (ID 3# for Text2Re and ID 4# for Re2Text, detailed in Table 2) are employed as baselines. The arrangements for the example variants are thoroughly detailed in Table 4. Within each group, we have three distinct sub-settings: 3-shot, 3-shot with hint & chain-of-thought (CoT, Wei et al. 2022b), and 6-shot.

number of triples for a particular relation is less than 20, we include all of them. Ultimately, the subset comprises a total of 328 triples. We run GPT-4 on both full set and subset in zero-shot settings. Results show that the subset can reflect the model's performance. Details can be found in Appendix D.

## 4 Results

In this section, we demonstrate the results of different LLM families on ConvRe benchmark and provide an in-depth analysis. More results on chat models can be found in Appendix E.

### 4.1 Converse Relation

Our first experiment, conducted in the zero-shot setting, involves both normal and converse relations across all model families. As shown in Figure 5, the performance on converse relations, within the scope of unaltered test text, is consistently inferior to that on normal relations across all tested models and tasks. More specifically, we note a roughly positive scaling trend for normal relations and an inverse scaling trend for converse relations, despite some outliers. The state-of-the-art LLM, GPT-4, underperforms compared to smaller models, with its performance falling significantly below random-guess levels. We conjecture that larger models have stronger priors, causing them to rely more heavily on memorized patterns from training data, which can conflict with the given task.

### 4.2 Text Variants

As introduced in Section 2.4, we are curious about LLMs' behaviours against text variants on the test and the few-shot examples.

Our initial focus is the zero-shot setting (Figure 5). For normal relations, test variants cause a noticeable performance drop. It means that if a given answer candidate fits the superficial pattern stated in the instruction, models are more likely to select it although it could be incorrect. This suggests that LLMs tend to take shortcut learning even within conventional problem settings. For converse relations, variants on the test text harm the performance on Re2Text while enhance it on Text2Re.

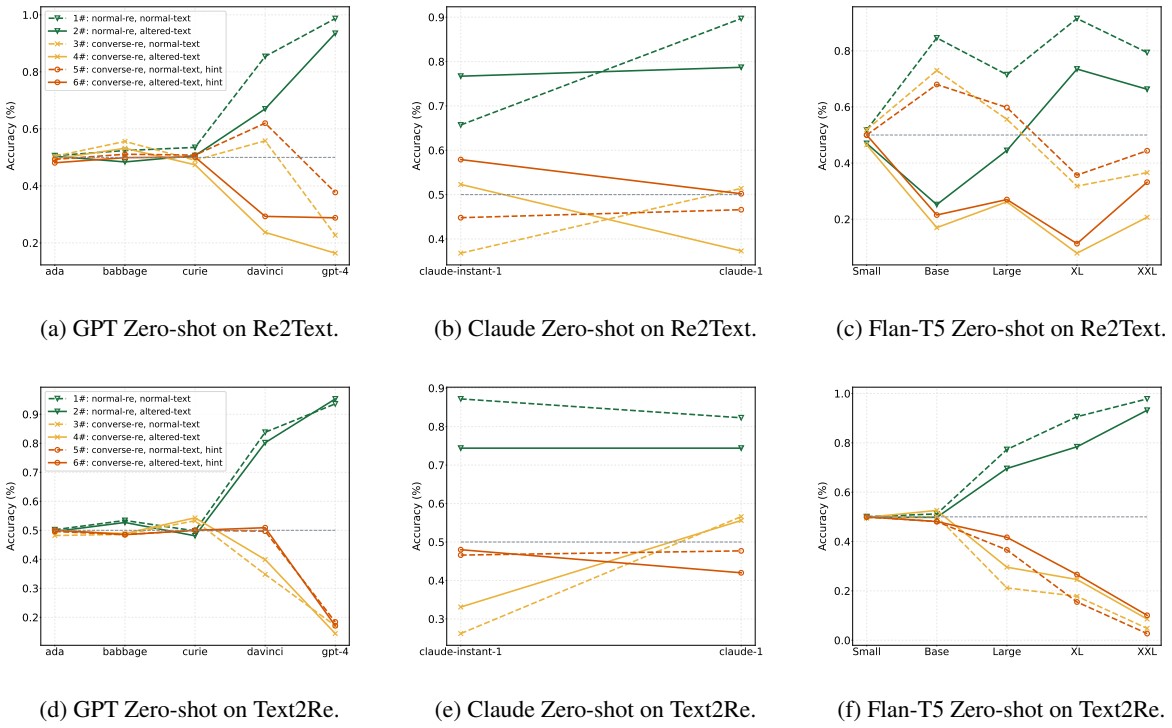

(a) GPT Zero-shot on Re2Text.

(b) Claude Zero-shot on Re2Text.

(c) Flan-T5 Zero-shot on Re2Text.

(d) GPT Zero-shot on Text2Re.

(e) Claude Zero-shot on Text2Re.

(f) Flan-T5 Zero-shot on Text2Re.

Figure 5: Zero-shot results on ConvRe. Each experimental setting has been indexed with a unique ID that can be referred to in Table 2. Sub-figures in the same row share the same figure legend, so we only display it once in the leftmost sub-figure to save space. The table version of the results can be found in Appendix I.

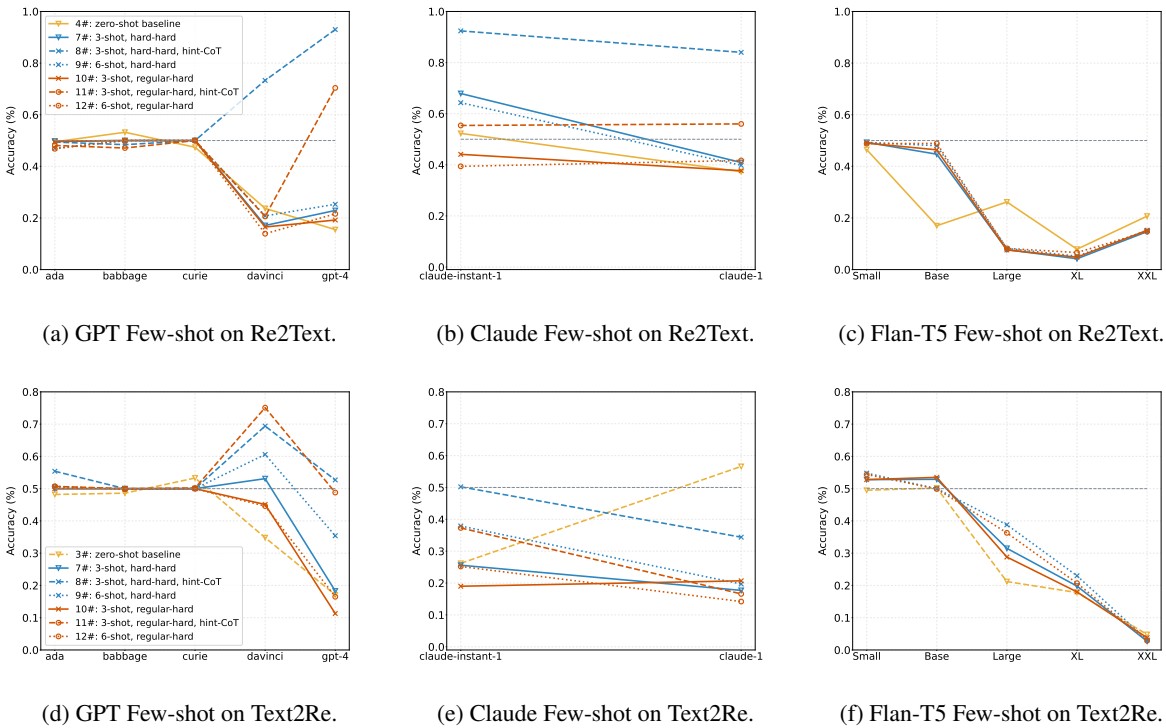

(a) GPT Few-shot on Re2Text.

(b) Claude Few-shot on Re2Text.

(c) Flan-T5 Few-shot on Re2Text.

(d) GPT Few-shot on Text2Re.

(e) Claude Few-shot on Text2Re.

(f) Flan-T5 Few-shot on Text2Re.

Figure 6: Few-shot results on ConvRe. Each experimental setting has been indexed with a unique ID that can be referred to in Table 3. Sub-figures in the same row share the same figure legend, so we only display it once in the leftmost sub-figure to save space. Detailed settings on the text variants can be found in Table 4. For GPT-4, we only test it on a subset of our benchmark. Due to Flan-T5's weak ability to follow CoT instructions, we do not report the results of Flan-T5 with hint and CoT prompting.

These findings lend strong support to our previous hypothesis presented in Section 2.4.

In the few-shot setting, the zero-shot baselines for both tasks are set to be *hard* (see Table 3 and 4). Generally, *hard* examples outperform *standard* examples (hard-hard vs. regular-hard) on average across different models on the two tasks. This can be attributed to the fact that *hard* examples align more consistently with the *hard* tests and effectively help models in avoiding bias and shortcut learning.

### 4.3 Shot Number

Examples, particularly an increased number of examples, are expected to outperform zero-shot prompting. However, we do not consistently observe improvements across different models and tasks. Notably, GPT models demonstrate the most consistent improvements, indicating superior in-context learning abilities among these models. Interestingly, when using few-shot examples, the models mostly exhibit inverse scaling or inverted U-shaped scaling, which suggests that our benchmark presents a challenge for the current LLMs.

### 4.4 Hint and CoT

The zero-shot experiments in Figure 5 indicate that the use of hints in prompts typically yields improvements for GPT and Flan-T5 models. However, `claude-1` stands out as an exception, appearing to be negatively affected by the hint.

In the few-shot experiments, employing hints and the Chain-of-Thought (CoT) approach substantially boosts performance, particularly for larger models. GPT models exhibit positive scaling and U-shaped scaling on the Re2Text task. However, for the Text2Re task, we still observe inverted U-shaped scaling for GPT models and inverse scaling for Claude models. This indicates that LLMs still struggle on this task even with strong prompting methods. We also find that Flan-T5 cannot properly follow CoT instructions, so we do not report the results of Flan-T5 with hint and CoT prompting.

### 5 Related Work

Studies on LLMs have shown positive scaling trends, whereby larger models generally perform better on downstream tasks (Brown et al., 2020; Rae et al., 2021; Chowdhery et al., 2022; Srivastava et al., 2022; Liang et al., 2022). However, researchers showed that model performance scal-

ing can deviate from naive expectations. Srivastava et al. (2022) showed slower and less smooth trends, and that social biases sometimes scale inversely with model size, a finding that is echoed in Parrish et al. (2022). TruthfulQA (Lin et al., 2022) demonstrated that while larger models can provide more informative answers, they tend to be less truthful. McKenzie et al. (2023) introduced the inverse scaling challenge and collected tasks that are highly atypical but still easily understandable by a human. Wei et al. (2022a) uncovered the U-shaped scaling trend by expanding the model scope for evaluation. Zhang et al. (2023) proposed NeQA and showed that this task exhibit inverse, U-shaped, or positive scaling with different prompt methods or model families. Miceli-Barone et al. (2023) showed that LLMs fail to correctly generate Python code when default identifiers are swapped.

Recent research has highlighted the issue of inflated performance in LLMs. Geirhos et al. (2020) coined the term shortcut learning, revealing models' reliance on superficial cues. Tu et al. (2020) studied the model's robustness to spurious correlations, which refers to the prediction rules that work for the majority examples but do not hold in general. Li et al. (2023b) found that LLMs tend to rely on shallow matching rather than understanding mathematical concepts. Bender et al. (2021) highlighted the importance of understanding the mechanism by which LLMs achieved state-of-the-art performance. Perez et al. (2021) showed that LLMs' few-shot ability is often overestimated due to the use of large held-out sets. Ji et al. (2023) surveyed the hallucination problem in language generation, highlighting the issue of factually incorrect output. Liu et al. (2023) identified attention glitches in Transformers, indicating a failure in capturing robust reasoning. Berglund et al. (2023) introduced the term reverse curse, showing that LLMs trained on 'A is B' fails to learn the reverse relationship 'B is A'.

### 6 Concolusion

In this paper, we present an investigation into LLMs' understanding of structured semantics, specifically focusing on converse binary relations. By introducing a novel benchmark, ConvRe, we offer a systematic and comprehensive evaluation suite to observe the performance of LLMs across diverse settings and prompting methods. We have carried out a detailed experimental study and observed various scaling trends that shed light on the

capabilities and limitations of LLMs. Our findings suggest that LLMs often resort to shortcut learning and still face considerable challenges on our proposed benchmark, even when strong prompting techniques are employed. Our work underscores the importance of developing evaluation methodologies to improve the understanding of LLMs and their performance across various tasks.

## Limitations

This paper proposes a new benchmark ConvRe to evaluate the competence of LLMs in recognizing and processing converse relations. Due to the limitation of the budget, we have evaluated three representative LLMs families on the subset of our benchmark for some settings. We note that the LLM APIs may change over time. Although we have set the sampling temperature to 0, we cannot fully guarantee the reproducibility of our results. Another potential limitation is the prompting methods used in this work. To automatically evaluate the model's performance, we have followed the previous studies and formatted the tasks as multiple-choice question answering tests. This setting may affect the performance of smaller models. Finally, due to the unknown data sources and pretraining methods used for proprietary models (e.g., Claude and GPT), it's difficult to arrive at a clear and comprehensive understanding of the behaviors exhibited by LLMs on our benchmark.

## Ethics Statement

Our work proposes a new benchmark to help reveal the real capability of LLMs in formal language oriented tasks. The triples in our benchmark are all extracted from publicly available and widely used knowledge graph dataset. We show that LLMs have taken shortcut learning in these tasks and their performance could be inflated. These findings may help users have a better understanding of LLMs and avoid the potential risks.

## Acknowledgements

This work is supported by the National Key Research and Development Program of China (Grant No. 2021YFB3300400) and National Natural Science Foundation of China (Grant No. 62173017).

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

## A Clarification about Similar Terminologies

As described in Tu et al. (2020), *spurious correlation* refers to the prediction rules that work for the majority examples but do not hold in general. *Superficial cues/biases/artifacts* can be treated as unintended correlations between input and output in existing datasets, which are often introduced during data collection or human annotation (Bender et al., 2021; Le Bras et al., 2020; Niven and Kao, 2019). The *shortcut* used in this paper refers to decision rules that perform well on standard benchmarks but fail to transfer to more challenging testing conditions, such as real word scenarios (Geirhos et al., 2020).

While these terms may have nuanced differences, their essence converges to the idea that models might exploit unintended patterns in datasets, particularly those evident in the majority of examples. This can harm their ability to generalize in open-world scenarios. In this paper, we have introduced

textual variance in our benchmark to serve as adversarial test sets and incorporated the counterfactual assumption to assess the real task-level generalization capabilities of LLMs.

## B Benchmark Details

To meet the second condition for relations in Sec 2.5, we merge the relation mother of person from NELL-ONE dataset with the relation father from Wikidata5M to create a new relation called parent of. In this way, there are 17 relations in total, and the detailed number of triples for each relation is shown in Table 5. The source knowledge graphs these relations come from cover a wide range of domains, such as socio-political and commonsense, which can ensure the diverseity of our dataset.

## C Model Family Details

### C.1 OpenAI GPT

The models we use in our experiments are mainly GPT-3 models (text-ada-001, text-babbage-001 and text-curie-001), GPT-3.5 models (text-davinci-003 and gpt-3.5-turbo) and GPT4. GPT-3 models can understand and generate natural language. These models were superceded by the more powerful GPT-3.5 generation models. Among the GPT-3.5 models, gpt-3.5-turbo has been optimized for chat but also works well for traditional completion tasks. The version of gpt-3.5-turbo we use in our experiments is gpt-3.5-turbo-0301. GPT-4 is a large multimodal model that can solve difficult problems with greater accuracy than any of the models in OpenAI GPT family, and the version we use for our experiments is gpt-4-0314.

### C.2 Anthropic Claude

Claude is capable of a wide variety of conversational and text processing tasks, it can help with use cases including summarization, search, creative and collaborative writing. Claude comes with two different sizes: claude-1 and claude-instant-1. claude-1 is the largest model in Claude family and ideal for a wide range of complex tasks. claude-instant-1 is a smaller model with far lower latency. Both of the models are provided with many different sub-versions. Among them, claude-1.3 and claude-instant-1.1 are used for our experiments.

| Relation | Numbers | Source | KG |
|---|---|---|---|
| hypernym | 80 | WN18RR | WordNet |
| has part | 78 | WN18RR | WordNet |
| organization, organization relationship, child | 75 | FB15K-237 | FreeBase |
| location, location, partially contains | 77 | FB15K-237 | FreeBase |
| athlete beat athlete | 80 | NELL-ONE | NELL |
| parent of (mother) | 145 | NELL-ONE | NELL |
| parent of (father) | | Wikidata5M | WikiData |
| represented by | 79 | Wikidata5M | WikiData |
| side effect | 8 | Wikidata5M | WikiData |
| has facility | 62 | Wikidata5M | WikiData |
| influenced by | 65 | Wikidata5M | WikiData |
| owned by | 51 | Wikidata5M | WikiData |
| consult | 73 | ICEWS14 | ICEWS |
| praise or endorse | 78 | ICEWS14 | ICEWS |
| made of | 80 | ConceptNet5 | ConceptNet |
| used of | 79 | ConceptNet5 | ConceptNet |
| has property | 55 | ConceptNet5 | ConceptNet |
| has subevent | 75 | ConceptNet5 | ConceptNet |
| Total | 1240 | | |

Table 5: The details of the relations in our ConvRe benchmark

## C.3 Google Flan-T5

Flan-T5 is an enhanced version of T5 that has been finetuned in a mixture of tasks. Unlike the OpenAI GPT model, Flan-T5 is an encoder-decoder model. There are five models with different sizes in Flan-T5 family: Flan-T5-Small, Flan-T5-Base, Flan-T5-Large, Flan-T5-XL and Flan-T5-XXL. All five models are used in our experiments.

## D Subset Results

To verify that the constructed subset can unbiasedly reflect the performance of GPT-4 model, we compare the performance of GPT-4 model on both benchmark dataset and subset. The results are shown in Table 6. The performance of the GPT-4 model shows minimal differences between the complete set and the subset, confirming the validity of the subset.

| Dataset | Prompt 1# | Prompt 2# | Prompt 3# | Prompt 4# |
|---|---|---|---|---|
| *Re2Text* | | | | |
| *complete set (1240)* | 0.987 | 0.935 | 0.227 | 0.164 |
| *subset (328)* | 0.997 | 0.942 | 0.192 | 0.155 |
| *Text2Re* | | | | |
| *complete set (1240)* | 0.936 | 0.953 | 0.171 | 0.144 |
| *subset (328)* | 0.942 | 0.951 | 0.171 | 0.165 |

Table 6: The comparison results of GPT-4 model on the complete set and subset under zero shot settings.

## E Chat Model Performance

As chat models usually have a better ability to follow instructions, they may demonstrate a different scaling trend on our benchmark. Therefore, we independently evaluate and compare the two chat model families (i.e. OpenAI GPT and Anthropic Claude) on our benchmark. As GPT-4 is also optimized for chat, we include it for analysis as well. The performances of the two families are shown in Figure 7.

In Re2Text task, it can be observed that few-shot with Chain-of-Thought can significantly improve the performance of GPT models. The accuracy of GPT4 demonstrates a remarkable improvement, soaring from below 0.2 in the zero-shot setting to surpassing 0.9 in the Few-shot+Hint+CoT setting. Chain-of-Thought is also helpful in improving the performance of Claude-1.

In Text2Re task, GPT models exhibit a distinct and consistent inverse scaling trend in both zero-shot and few-shot settings when the relation is conversed. However, the scaling trend of Claude models is more intricate. Specifically, in zero-shot settings, Claude models demonstrate a positive scaling trend in the majority of settings. In few-shot settings, on the contrary, an inverse scaling trend is exhibited by Claude models.

## F Model Behaviors

This section introduces the behaviors of different models that we observe during experiments. Under zero-shot settings, Claude and Flan-T5 can

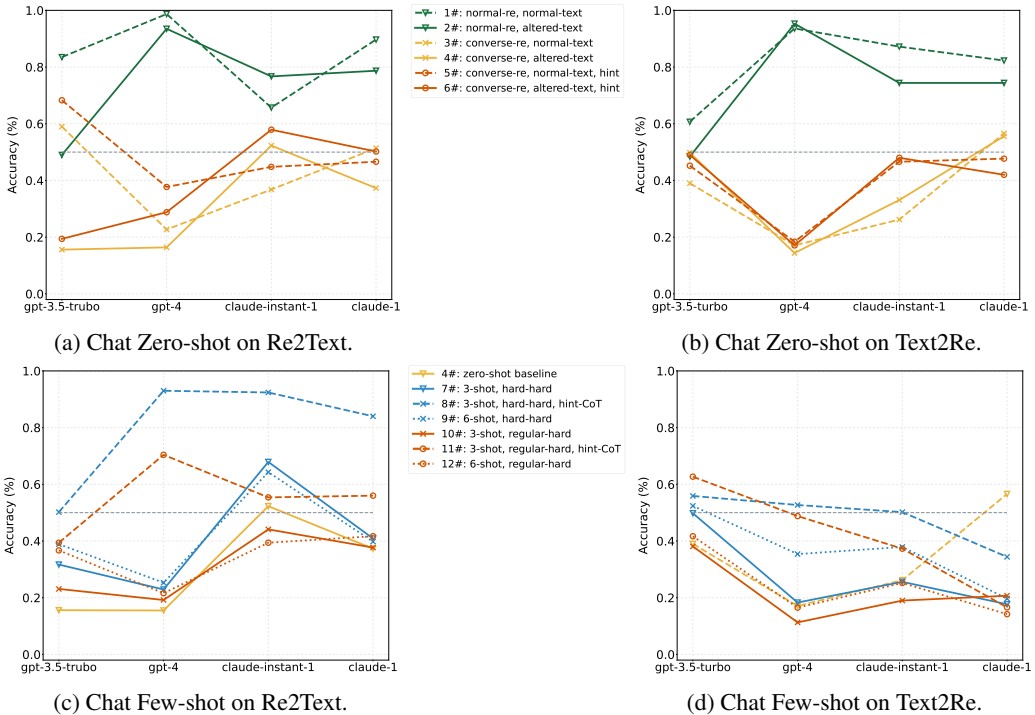

(a) Chat Zero-shot on Re2Text.

(b) Chat Zero-shot on Text2Re.

(c) Chat Few-shot on Re2Text.

(d) Chat Few-shot on Text2Re.

Figure 7: Zero-shot and Few-shot results of chat models on ConvRe. Each experimental setting has been indexed with a unique ID that can be referred in Table 2. Sub-figures in the same row share the centeral figure legend.

generate answers in the expected behavior. However, text-ada-001 and text-babbage-001 fails in most cases, they tend to repeat our question or instruction. In our experiments, if these two models don't give a clear answer, we will treat the choices with higher log probability in the first token as their answers. In few-shot settings, nearly all models except Flan-T5 conform to the expected answer format. The generated thoughts of Flan-T5 are usually shorter than the examples, and the format of its answer seldom aligns with the expected format.

## G Neutral Relation Results

In this section, we explore the impact of neutral relations on ConvRe benchmark. Specifically, we change the relation text to a more neutral name: relation R, and then run the experiments on the subset mentioned in Appendix D. The results are shown in Table 7.

It can be observed that altering symbols to adopt more neutral names generally shows various effects on the models. The performance of most models in prompt 3# and 4# (the challenging setup) is still around 50% or even worse. However, for Claude models, considerable improvements on converse relations (prompt 3#, 4#, 7# and 8#) can be observed in the Text2Re task, along with the performance

drop on normal relations (prompt 1# and 2#).

In conclusion, altering relation text to more neutral forms may help alleviate problems in understanding converse relations, but it carries the risk of harming the performance in normal relations.

## H Analysis of the Impact of Different Entity Pairs

We firstly extract all the triples with relation hypernym and run five different models on them within the hard setting (prompt 4#) of Re2Text task. Then the overlap percentages of the wrongly answered triples across the models are calculated. The results are shown in Table 8. The diverse accuracies and low overlap percentages of incorrectly answered entity pairs indicate that different entity pairs for the same relation indeed lead to different results on different models.

## I The Table Version of the Results

We provide the table of our entire experimental results in Table 9 for better clarity.

## J Prompt Examples

Figure 8 to Figure 19 demonstrate the 12 kinds of prompts used in Re2Text tasks.

| Model | Prompt 1# | Prompt 2# | Prompt 3# | Prompt 4# | Prompt 7# | Prompt 8# |
|---|---|---|---|---|---|---|
| | | | *Re2Text* | | | |
| text-ada-001 | 0.494 (-0.040) | 0.509 (-0.003) | 0.509 (-0.003) | 0.515 (+0.015) | 0.494 (-0.012) | 0.5 (+0.003) |
| text-babbage-001 | 0.518 (+0.051) | 0.537 (+0.074) | 0.537 (-0.015) | 0.527 (+0.012) | 0.500 (-0.009) | 0.466 (-0.019) |
| text-curie-001 | 0.527 (+0.006) | 0.463 (-0.031) | 0.448 (-0.046) | 0.439 (-0.037) | 0.500 (-0.009) | 0.500 (-0.009) |
| text-davinci-003 | 0.857 (-0.006) | 0.567 (-0.134) | 0.659 (+0.074) | 0.259 (+0.027) | 0.101 (-0.054) | 0.774 (+0.094) |
| gpt-3.5-turbo | 0.765 (-0.073) | 0.616 (+0.134) | 0.384 (-0.211) | 0.229 (+0.083) | 0.439 (+0.110) | 0.716 (+0.253) |
| gpt-4 | 0.985 (-0.003) | 0.918 (-0.027) | 0.561 (+0.335) | 0.439 (+0.268) | 0.317 (+0.088) | 0.784 (-0.146) |
| claude-1 | 0.905 (+0.003) | 0.777 (-0.031) | 0.732 (+0.198) | 0.537 (+0.165) | 0.335 (-0.055) | 0.848 (+0.031) |
| claude-instant-1 | 0.762 (+0.073) | 0.613 (-0.152) | 0.485 (+0.113) | 0.384 (-0.122) | 0.558 (-0.104) | 0.777 (-0.144) |
| flan-t5-small | 0.546 (+0.025) | 0.488 (+0.015) | 0.546 (+0.006) | 0.494 (+0.046) | 0.506 (+0.027) | - |
| flan-t5-base | 0.796 (-0.042) | 0.329 (+0.061) | 0.665 (-0.039) | 0.201 (+0.049) | 0.488 (+0.049) | - |
| flan-t5-large | 0.634 (-0.061) | 0.430 (-0.012) | 0.558 (+0.003) | 0.378 (+0.094) | 0.253 (+0.146) | - |
| flan-t5-xl | 0.875 (-0.046) | 0.546 (-0.201) | 0.518 (+0.216) | 0.210 (+0.131) | 0.183 (+0.137) | - |
| flan-t5-xxl | 0.738 (-0.070) | 0.591 (-0.095) | 0.476 (+0.104) | 0.290 (+0.064) | 0.180 (+0.034) | - |
| | | | *Text2Re* | | | |
| text-ada-001 | 0.494 (-0.040) | 0.509 (-0.003) | 0.509 (-0.003) | 0.515 (+0.015) | 0.494 (-0.012) | 0.5 (+0.003) |
| text-babbage-001 | 0.518 (+0.051) | 0.537 (+0.074) | 0.537 (-0.015) | 0.527 (+0.012) | 0.500 (-0.009) | 0.466 (-0.019) |
| text-curie-001 | 0.527 (+0.006) | 0.463 (-0.031) | 0.448 (-0.046) | 0.439 (-0.037) | 0.500 (-0.009) | 0.500 (-0.009) |
| text-davinci-003 | 0.857 (-0.006) | 0.567 (-0.134) | 0.659 (+0.074) | 0.259 (+0.027) | 0.101 (-0.054) | 0.774 (+0.094) |
| gpt-3.5-turbo | 0.765 (-0.073) | 0.616 (+0.134) | 0.384 (-0.211) | 0.229 (+0.083) | 0.439 (+0.110) | 0.716 (+0.253) |
| gpt-4 | 0.985 (-0.003) | 0.918 (-0.027) | 0.561 (+0.335) | 0.439 (+0.268) | 0.317 (+0.088) | 0.784 (-0.146) |
| claude-1 | 0.905 (+0.003) | 0.777 (-0.031) | 0.732 (+0.198) | 0.537 (+0.165) | 0.335 (-0.055) | 0.848 (+0.031) |
| claude-instant-1 | 0.762 (+0.073) | 0.613 (-0.152) | 0.485 (+0.113) | 0.384 (-0.122) | 0.558 (-0.104) | 0.777 (-0.144) |
| flan-t5-small | 0.546 (+0.025) | 0.488 (+0.015) | 0.546 (+0.006) | 0.494 (+0.046) | 0.506 (+0.027) | - |
| flan-t5-base | 0.796 (-0.042) | 0.329 (+0.061) | 0.665 (-0.039) | 0.201 (+0.049) | 0.488 (+0.049) | - |
| flan-t5-large | 0.634 (-0.061) | 0.430 (-0.012) | 0.558 (+0.003) | 0.378 (+0.094) | 0.253 (+0.146) | - |
| flan-t5-xl | 0.875 (-0.046) | 0.546 (-0.201) | 0.518 (+0.216) | 0.210 (+0.131) | 0.183 (+0.137) | - |
| flan-t5-xxl | 0.738 (-0.070) | 0.591 (-0.095) | 0.476 (+0.104) | 0.290 (+0.064) | 0.180 (+0.034) | - |

Table 7: The performance of LLMs on ConvRe benchmark after altering relation text to `relation R`. The number in the parentheses represents the difference between the neutral relation naming and normal naming under the same setup on the same subset. We do not report the results of Flan-T5 as it struggles to follow the Chain-of-Thought instructions.

| | gpt-3.5-turbo | gpt-4 | claude-1 | claude-instant-1 | flan-t5-xxl |
|---|---|---|---|---|---|
| *Accuracy* | 77.50% | 71.25% | 100.00% | 83.75% | 47.50% |
| | | *overlap percentage of incorrectly answered entity pairs* | | | |
| gpt-3.5-turbo | - | 11.25% | 0.00% | 1.25% | 17.50% |
| gpt-4 | 11.25% | - | 0.00% | 2.50% | 25.00% |
| claude-1 | 0.00% | 0.00% | - | 0.00% | 0.00% |
| claude-instant-1 | 1.25% | 2.50% | 0.00% | - | 0.00% |
| flan-t5-xxl | 17.50% | 25.00% | 0.00% | 0.00% | - |

Table 8: The accuracy of five different models on relation `hypernym` in the hard setting (prompt 4#) of Re2Text task. The bottom part shows the overlap percentage of incorrectly answered entity pairs between these models.

| Model | Prompt 1# | Prompt 2# | Prompt 3# | Prompt 4# | Prompt 5# | Prompt 6# | Prompt 7# | Prompt 8# | Prompt 9# | Prompt 10# | Prompt 11# | Prompt 12# |
|---|---|---|---|---|---|---|---|---|---|---|---|---|
| | | | | | | *Re2Text* | | | | | | | |
| text-ada-001 | 0.506 | 0.504 | 0.504 | 0.495 | 0.493 | 0.481 | 0.498 | 0.494 | 0.470 | 0.496 | 0.481 | 0.468 |
| text-babbage-001 | 0.524 | 0.484 | 0.556 | 0.532 | 0.511 | 0.499 | 0.500 | 0.484 | 0.500 | 0.500 | 0.471 | 0.500 |
| text-curie-001 | 0.535 | 0.506 | 0.490 | 0.474 | 0.508 | 0.501 | 0.500 | 0.500 | 0.500 | 0.500 | 0.500 | 0.500 |
| text-davinci-003 | 0.854 | 0.670 | 0.558 | 0.237 | 0.620 | 0.293 | 0.171 | 0.733 | 0.208 | 0.165 | 0.206 | 0.139 |
| gpt-3.5-turbo | 0.835 | 0.490 | 0.590 | 0.156 | 0.683 | 0.194 | 0.317 | 0.502 | 0.389 | 0.231 | 0.394 | 0.367 |
| gpt-4 | 0.987 | 0.935 | 0.227 | 0.164 | 0.377 | 0.288 | 0.229 | 0.930 | 0.253 | 0.192 | 0.704 | 0.216 |
| claude-instant-1 | 0.657 | 0.767 | 0.368 | 0.523 | 0.448 | 0.579 | 0.679 | 0.924 | 0.643 | 0.441 | 0.554 | 0.394 |
| claude-1 | 0.897 | 0.787 | 0.514 | 0.373 | 0.466 | 0.502 | 0.409 | 0.840 | 0.398 | 0.377 | 0.560 | 0.417 |
| flan-t5-small | 0.518 | 0.470 | 0.519 | 0.465 | 0.500 | 0.500 | 0.493 | - | 0.493 | 0.490 | - | 0.488 |
| flan-t5-base | 0.846 | 0.252 | 0.730 | 0.170 | 0.680 | 0.215 | 0.447 | - | 0.480 | 0.464 | - | 0.489 |
| flan-t5-large | 0.715 | 0.445 | 0.556 | 0.262 | 0.598 | 0.270 | 0.077 | - | 0.082 | 0.076 | - | 0.082 |
| flan-t5-xl | 0.915 | 0.735 | 0.318 | 0.079 | 0.357 | 0.113 | 0.042 | - | 0.052 | 0.048 | - | 0.066 |
| flan-t5-xxl | 0.794 | 0.663 | 0.366 | 0.207 | 0.444 | 0.332 | 0.147 | - | 0.152 | 0.152 | - | 0.147 |
| | | | | | | *Text2Re* | | | | | | | |
| text-ada-001 | 0.502 | 0.496 | 0.482 | 0.498 | 0.496 | 0.501 | 0.500 | 0.554 | 0.506 | 0.500 | 0.507 | 0.507 |
| text-babbage-001 | 0.534 | 0.527 | 0.486 | 0.489 | 0.485 | 0.485 | 0.500 | 0.500 | 0.499 | 0.500 | 0.500 | 0.498 |
| text-curie-001 | 0.498 | 0.481 | 0.533 | 0.543 | 0.500 | 0.500 | 0.500 | 0.500 | 0.500 | 0.500 | 0.502 | 0.500 |
| text-davinci-003 | 0.838 | 0.802 | 0.348 | 0.399 | 0.497 | 0.509 | 0.531 | 0.694 | 0.606 | 0.451 | 0.751 | 0.446 |
| gpt-3.5-turbo | 0.607 | 0.484 | 0.390 | 0.498 | 0.452 | 0.490 | 0.498 | 0.559 | 0.524 | 0.381 | 0.627 | 0.417 |
| gpt-4 | 0.936 | 0.953 | 0.171 | 0.144 | 0.184 | 0.171 | 0.183 | 0.527 | 0.354 | 0.113 | 0.488 | 0.165 |
| claude-instant-1 | 0.872 | 0.744 | 0.262 | 0.331 | 0.466 | 0.480 | 0.256 | 0.502 | 0.379 | 0.190 | 0.373 | 0.252 |
| claude-1 | 0.823 | 0.744 | 0.566 | 0.556 | 0.477 | 0.420 | 0.177 | 0.344 | 0.197 | 0.207 | 0.166 | 0.142 |
| flan-t5-small | 0.501 | 0.498 | 0.495 | 0.498 | 0.500 | 0.500 | 0.528 | - | 0.548 | 0.528 | - | 0.544 |
| flan-t5-base | 0.512 | 0.498 | 0.502 | 0.526 | 0.481 | 0.481 | 0.529 | - | 0.500 | 0.535 | - | 0.499 |
| flan-t5-large | 0.773 | 0.696 | 0.212 | 0.296 | 0.366 | 0.417 | 0.315 | - | 0.388 | 0.288 | - | 0.363 |
| flan-t5-xl | 0.906 | 0.784 | 0.178 | 0.246 | 0.155 | 0.266 | 0.198 | - | 0.230 | 0.180 | - | 0.207 |
| flan-t5-xxl | 0.978 | 0.932 | 0.048 | 0.086 | 0.027 | 0.101 | 0.027 | - | 0.031 | 0.038 | - | 0.029 |

Table 9: The table of our entire experiments.

---

**Prompt**

Read the instruction and then answer the question using A or B.

- - - - - - - - - - - - - - - - - - - - - - - - - - - - - - - - - - - - - - - - - - - - - - - - - - - - - - - - - - - - - - - - - - - - - - - - - - - - - - - - - - - - - - - - - -

Instruction: (x, has part, y) indicates that x has a part called y.
Question: (?, has part, solingen)
A: Find an entity that solingen contains.
B: Find an entity that has a part called solingen.
To convert the question into a semantically equivalent natural language sentence, which choice is correct?
Answer:
Expected Answer: B

Figure 8: Prompt Design 1#

---

**Prompt**

Read the instruction and then answer the question using A or B.

- - - - - - - - - - - - - - - - - - - - - - - - - - - - - - - - - - - - - - - - - - - - - - - - - - - - - - - - - - - - - - - - - - - - - - - - - - - - - - - - - - - - - - - - - -

Instruction: (x, has part, y) indicates that x has a part called y.
Question: (?, has part, solingen)
A: Find an entity that is a part of solingen.
B: Find an entity that possesses a specific component named solingen.
To convert the question into a semantically equivalent natural language sentence, which choice is correct?
Answer:
Expected Answer: B

Figure 9: Prompt Design 2#

**Prompt**

Read the instruction and then answer the question using A or B.

--------------------------------------------------------------------------------------------------------------------

Instruction: (x, has part, y) indicates that y has a part called x.
Question: (?, has part, solingen)
A: Find an entity that possesses a specific component named solingen.
B: Find an entity that is a part of solingen.
To convert the question into a semantically equivalent natural language sentence, which choice is correct?
Answer:
Expected Answer: B

Figure 10: Prompt Design 3#

**Prompt**

Read the instruction and then answer the question using A or B.

--------------------------------------------------------------------------------------------------------------------

Instruction: (x, has part, y) indicates that y has a part called x.
Question: (?, has part, solingen)
A: Find an entity that has a part called solingen.
B: Find an entity that solingen contains.
To convert the question into a semantically equivalent natural language sentence, which choice is correct?
Answer:
Expected Answer: B

Figure 11: Prompt Design 4#

**Prompt**

Read the instruction and then answer the question using A or B. Note that in this task, if the relation is defined in a converse manner, unlike the conventional definition, you should carefully choose the answer.

--------------------------------------------------------------------------------------------------------------------

Instruction: (x, has part, y) indicates that y has a part called x.
Question: (?, has part, solingen)
A: Find an entity that possesses a specific component named solingen.
B: Find an entity that is a part of solingen.
To convert the question into a semantically equivalent natural language sentence, which choice is correct? Look out for the ORDER of the entities in the instruction!
Answer:
Expected Answer: B

Figure 12: Prompt Design 5#

**Prompt**

Read the instruction and then answer the question using A or B. Note that in this task, if the relation is defined in a converse manner, unlike the conventional definition, you should carefully choose the answer.

--------------------------------------------------------------------------------------------------------------------

Instruction: (x, has part, y) indicates that y has a part called x.
Question: (?, has part, solingen)
A: Find an entity that has a part called solingen.
B: Find an entity that solingen contains.
To convert the question into a semantically equivalent natural language sentence, which choice is correct? Look out for the ORDER of the entities in the instruction!
Answer:
Expected Answer: B

Figure 13: Prompt Design 6#

| Prompt |
|---|
| Read the instruction and then answer the question using A or B.
-----------------------------------------------------------------------------------------------------------------------------
[Example1]
Instruction: (x, works for, y) indicates that y works for x.
Question: (?, works for, anthony fauci)
A: Find an entity that works for anthony fauci.
B: Find an entity that anthony fauci is employed by.
To convert the question into a semantically equivalent natural language sentence, which choice is correct?
Answer: B
-----------------------------------------------------------------------------------------------------------------------------
[Example2]
Instruction: (x, bigger than, y) indicates that y is bigger than x.
Question: (?, bigger than, elephant)
A: Find an entity that is smaller than elephant.
B: Find an entity that is bigger than elephant.
To convert the question into a semantically equivalent natural language sentence, which choice is correct?
Answer: A
-----------------------------------------------------------------------------------------------------------------------------
[Example3]
Instruction: (x, in the south of, y) indicates that y is in the south of x.
Question: (?, in the south of, china)
A: Find an entity that is in the north of china.
B: Find an entity that is in the south of china.
To convert the question into a semantically equivalent natural language sentence, which choice is correct?
Answer: A
-----------------------------------------------------------------------------------------------------------------------------
Instruction: (x, has part, y) indicates that y has a part called x.
Question: (?, has part, solingen)
A: Find an entity that has a part called solingen.
B: Find an entity that solingen contains.
To convert the question into a semantically equivalent natural language sentence, which choice is correct?
Answer:
Expected Answer: B |

Figure 14: Prompt Design 7#

**Prompt**

Read the instruction and then answer the question using A or B. Note that in this task, if the relation is defined in a converse manner, unlike the conventional definition, you should carefully choose the answer. Your answer should be in JSON format with the following keys: thought, answer.

-----------------------------------------------------------------------------------------------------------------

[Example1]
Instruction: (x, works for, y) indicates that y works for x.
Question: (?, works for, anthony fauci)
A: Find an entity that works for anthony fauci.
B: Find an entity that anthony fauci is employed by.
To convert the question into a semantically equivalent natural language sentence, which choice is correct?
Answer: {'thought': "Let's think step by step. Firstly, the question is asking for x. Then, the instruction indicates y works for x. According to the question, y is anthony fauci, and therefore anthony fauci works for x. So x is the employer of anthony fauci, the answer is B.", 'answer': 'B'}

-----------------------------------------------------------------------------------------------------------------

[Example2]
Instruction: (x, bigger than, y) indicates that y is bigger than x.
Question: (?, bigger than, elephant)
A: Find an entity that is smaller than elephant.
B: Find an entity that is bigger than elephant.
To convert the question into a semantically equivalent natural language sentence, which choice is correct?
Answer: {'thought': "Let's think step by step. Firstly, the question is asking for x. Then, the instruction indicates y is bigger than x. According to the question, y is elephant, and therefore elephant is bigger than x. So x is smaller than elephant, the answer is A.", 'answer': 'A'}

-----------------------------------------------------------------------------------------------------------------

[Example3]
Instruction: (x, in the south of, y) indicates that y is in the south of x.
Question: (?, in the south of, china)
A: Find an entity that is in the north of china.
B: Find an entity that is in the south of china.
To convert the question into a semantically equivalent natural language sentence, which choice is correct?
Answer: {'thought': "Let's think step by step. Firstly, the question is asking for x. Then, the instruction indicates y is in the south of x. According to the question, y is china, and therefore china is in the south of x. So x is in the north of china, the answer is A.", 'answer': 'A'}

-----------------------------------------------------------------------------------------------------------------

Instruction: (x, has part, y) indicates that y has a part called x.
Question: (?, has part, solingen)
A: Find an entity that has a part called solingen.
B: Find an entity that solingen contains.
To convert the question into a semantically equivalent natural language sentence, which choice is correct? Look out for the ORDER of the entities in the instruction!
Answer:
Expected Answer: B

Figure 15: Prompt Design 8#

**Prompt**

Read the instruction and then answer the question using A or B.

------------------------------------------------------------------------------------------------------

[Example1]
Instruction: (x, works for, y) indicates that y works for x.
Question: (?, works for, anthony fauci)
A: Find an entity that works for anthony fauci.
B: Find an entity that anthony fauci is employed by.
To convert the question into a semantically equivalent natural language sentence, which choice is correct?
Answer: B

------------------------------------------------------------------------------------------------------

[Example2]
Instruction: (x, bigger than, y) indicates that y is bigger than x.
Question: (?, bigger than, elephant)
A: Find an entity that is smaller than elephant.
B: Find an entity that is bigger than elephant.
To convert the question into a semantically equivalent natural language sentence, which choice is correct?
Answer: A

------------------------------------------------------------------------------------------------------

[Example3]
Instruction: (x, in the south of, y) indicates that y is in the south of x.
Question: (?, in the south of, china)
A: Find an entity that is in the north of china.
B: Find an entity that is in the south of china.
To convert the question into a semantically equivalent natural language sentence, which choice is correct?
Answer: A

------------------------------------------------------------------------------------------------------

[Example4]
Instruction: (x, teach, y) indicates that y teaches x.
Question: (?, teach, andy bramante)
A: Find a person that teaches andy bramante.
B: Find a person that is the student of andy bramante.
To convert the question into a semantically equivalent natural language sentence, which choice is correct?
Answer: B

------------------------------------------------------------------------------------------------------

[Example5]
Instruction: (x, interviewed, y) indicates that y interviewed x.
Question: (?, interviewed, biden)
A: Find a person that biden conducted an interviewed with.
B: Find a person that interviewed biden.
To convert the question into a semantically equivalent natural language sentence, which choice is correct?
Answer: A

------------------------------------------------------------------------------------------------------

[Example6]
Instruction: (x, successor, y) indicates that y is the successor of x.
Question: (?, successor, barack obama)
A: Find a person that is the successor of barack obama.
B: Find a person that is the predecessor of barack obama.
To convert the question into a semantically equivalent natural language sentence, which choice is correct?
Answer: B

------------------------------------------------------------------------------------------------------

Instruction: (x, has part, y) indicates that y has a part called x.
Question: (?, has part, solingen)
A: Find an entity that has a part called solingen.
B: Find an entity that solingen contains.
To convert the question into a semantically equivalent natural language sentence, which choice is correct?
Answer:
Expected Answer: B

Figure 16: Prompt Design 9#

| **Prompt** |
|---|
| Read the instruction and then answer the question using A or B. |

[Example1]
Instruction: (x, works for, y) indicates that y works for x.
Question: (?, works for, anthony fauci)
A: Find an entity that is employed by anthony fauci.
B: Find an entity that anthony fauci works for.
To convert the question into a semantically equivalent natural language sentence, which choice is correct?
Answer: B

[Example2]
Instruction: (x, bigger than, y) indicates that y is bigger than x.
Question: (?, bigger than, elephant)
A: Find an entity so that elephant is bigger than it.
B: Find an entity so that elephant is smaller than it.
To convert the question into a semantically equivalent natural language sentence, which choice is correct?
Answer: A

[Example3]
Instruction: (x, in the south of, y) indicates that y is in the south of x.
Question: (?, in the south of, china)
A: Find an entity so that china is in the south of it.
B: Find an entity so that china is in the north of it.
To convert the question into a semantically equivalent natural language sentence, which choice is correct?
Answer: A

Instruction: (x, has part, y) indicates that y has a part called x.
Question: (?, has part, solingen)
A: Find an entity that has a part called solingen.
B: Find an entity that solingen contains.
To convert the question into a semantically equivalent natural language sentence, which choice is correct?
Answer:
Expected Answer: B

Figure 17: Prompt Design 10#

**Prompt**

Read the instruction and then answer the question using A or B. Note that in this task, if the relation is defined in a converse manner, unlike the conventional definition, you should carefully choose the answer. Your answer should be in JSON format with the following keys: thought, answer.

----------------------------------------------------------------------------------------------------------------------------

[Example 1]
Instruction: (x, works for, y) indicates that y works for x.
Question: (?, works for, anthony fauci)
A: Find an entity that is employed by anthony fauci.
B: Find an entity that anthony fauci works for.
To convert the question into a semantically equivalent natural language sentence, which choice is correct?
Answer: {'thought': "Let's think step by step. Firstly, the question is asking for x. Then, the instruction indicates y works for x. According to the question, y is anthony fauci, and therefore anthony fauci works for x, the answer is B.", 'answer': 'B'}

----------------------------------------------------------------------------------------------------------------------------

[Example 2]
Instruction: (x, bigger than, y) indicates that y is bigger than x.
Question: (?, bigger than, elephant)
A: Find an entity so that elephant is bigger than it.
B: Find an entity so that elephant is smaller than it.
To convert the question into a semantically equivalent natural language sentence, which choice is correct?
Answer: {'thought': "Let's think step by step. Firstly, the question is asking for x. Then, the instruction indicates y is bigger than x. According to the question, y is elephant, and therefore elephant is bigger than x, the answer is A.", 'answer': 'A'}

----------------------------------------------------------------------------------------------------------------------------

[Example 3]
Instruction: (x, in the south of, y) indicates that y is in the south of x.
Question: (?, in the south of, china)
A: Find an entity so that china is in the south of it.
B: Find an entity so that china is in the north of it.
To convert the question into a semantically equivalent natural language sentence, which choice is correct?
Answer: {'thought': "Let's think step by step. Firstly, the question is asking for x. Then, the instruction indicates y is in the south of x. According to the question, y is china, and therefore china is in the south of x, the answer is A.", 'answer': 'A'}

----------------------------------------------------------------------------------------------------------------------------

Instruction: (x, has part, y) indicates that y has a part called x.
Question: (?, has part, solingen)
A: Find an entity that has a part called solingen.
B: Find an entity that solingen contains.
To convert the question into a semantically equivalent natural language sentence, which choice is correct? Look out for the ORDER of the entities in the instruction!
Answer:
Expected Answer: B

Figure 18: Prompt Design 11#

| **Prompt** |

Read the instruction and then answer the question using A or B.

------------------------------------------------------------------------------------------------------------------------

[Example1]
Instruction: (x, works for, y) indicates that y works for x.
Question: (?, works for, anthony fauci)
A: Find an entity that is employed by anthony fauci.
B: Find an entity that anthony fauci works for.
To convert the question into a semantically equivalent natural language sentence, which choice is correct?
Answer: B

------------------------------------------------------------------------------------------------------------------------

[Example2]
Instruction: (x, bigger than, y) indicates that y is bigger than x.
Question: (?, bigger than, elephant)
A: Find an entity so that elephant is bigger than it.
B: Find an entity so that elephant is smaller than it.
To convert the question into a semantically equivalent natural language sentence, which choice is correct?
Answer: A

------------------------------------------------------------------------------------------------------------------------

[Example3]
Instruction: (x, in the south of, y) indicates that y is in the south of x.
Question: (?, in the south of, china)
A: Find an entity so that china is in the south of it.
B: Find an entity so that china is in the north of it.
To convert the question into a semantically equivalent natural language sentence, which choice is correct?
Answer: A

------------------------------------------------------------------------------------------------------------------------

[Example4]
Instruction: (x, teach, y) indicates that y teaches x.
Question: (?, teach, andy bramante)
A: Find a person that andy bramante is the student of.
B: Find a person that andy bramante teaches.
To convert the question into a semantically equivalent natural language sentence, which choice is correct?
Answer: B

------------------------------------------------------------------------------------------------------------------------

[Example5]
Instruction: (x, interviewed, y) indicates that y interviewed x.
Question: (?, interviewed, biden)
A: Find a person that biden interviewed.
B: Find a person that conducted an interview with biden.
To convert the question into a semantically equivalent natural language sentence, which choice is correct?
Answer: A

------------------------------------------------------------------------------------------------------------------------

[Example6]
Instruction: (x, successor, y) indicates that y is the successor of x.
Question: (?, successor, barack obama)
A: Find a person that barack obama is the predecessor of.
B: Find a person that barack obama is the successor of.
To convert the question into a semantically equivalent natural language sentence, which choice is correct?
Answer: B

------------------------------------------------------------------------------------------------------------------------

Instruction: (x, has part, y) indicates that y has a part called x.
Question: (?, has part, solingen)
A: Find an entity that has a part called solingen.
B: Find an entity that solingen contains.
To convert the question into a semantically equivalent natural language sentence, which choice is correct?
Answer:
Expected Answer: B

Figure 19: Prompt Design 12#