# OpenReview forum: "An Investigation of LLMs’ Inefficacy in Understanding Converse Relations"
_EMNLP/2023/Conference — EMNLP 2023 Main_

### Official Review · Reviewer_2Eqb · 2023-07-28

**Typos Grammar Style And Presentation Improvements:** Figure 6 legends is too small to read…
**Soundness:** 3

**Excitement:**

4: Strong: This paper deepens the understanding of some phenomenon or lowers the barriers to an existing research direction.

**Paper Topic And Main Contributions:**

The authors of this paper introduce a benchmark for evaluating language models' ability to understand converse relations. Converse relations are relations where the order of the arguments is reversed. The instructions/prompts in the benchmark redefine the order of the arguments of a relation. E.g. a query (x, has part, y) gets a new interpretation which is instead of "x has part y" is "y has part x".

The benchmark is divided into two categories: Re2Text and Text2Re. In the Re2Text category, the model is given an instruction and a query, and it must choose a correct textual description of the query. In the Text2Re category, the model is given an instruction and a textual description of a query, and it must choose a correct query for that description.

The benchmark consists of 17 relations and 1240 triples. The evaluation is performed on three families of language models: GPT, CLAUDE, and FLAN-T5 with different sizes. The evaluation is done in a zero-shot setting, as well as with 3-shot and 6-shot training.

The authors show that the performance of language models on this benchmark decreases as the size of the model increases. They suggest that LLMs often resort to shortcut learning.

The main contribution of the paper is the benchmark and the evaluation

**Questions For The Authors:**

Question A: What would be human performance? The task is quite easy, but I'm not sure if the average human rater (not connected with academia) would have a 100% success rate.

Question B: Why, in the benchmark, every relation needs so many entity pairs? The entities don't influence how the question/instruction/prompt is phrased. Did you observe that different entities/pairs of entities for the same relation lead to different results?

Question C: What version of ChatGPT/GPT4 was used (from which date)?

**Reasons To Accept:**

The authors:
 - Provides a benchmark for understanding converse relations that poses a challenge for LLMs.
 - Conducts a thorough evaluation on the benchmark showing inverse scaling trend.
 - Quite clear presentation.


**Reasons To Reject:**

 - Not reproducible results on ChatGPT and GPT4 (however, fortunately there are results on the open sourced FLAN-T5).

**Reproducibility:**

2: Would be hard pressed to reproduce the results. The contribution depends on data that are simply not available outside the author's institution or consortium; not enough details are provided.

**Reviewer Confidence:**

3: Pretty sure, but there's a chance I missed something. Although I have a good feel for this area in general, I did not carefully check the paper's details, e.g., the math, experimental design, or novelty.

---

> ### Author Rebuttal · Authors · 2023-08-29
>
> Thank you for your detailed comments, and they are really helpful for us to improve our paper. We will carefully incorporate them in the revised paper.
>
> > Q1: Not reproducible results on ChatGPT and GPT4 (however, fortunately there are results on the open sourced FLAN-T5).
>
> Our results can be reproduced using the codebase in the original supplementary material, including ChatGPT and GPT4. Besides, we promise to release our codes to the public upon acceptance of the paper. We recognize the challenges that can arise due to potential changes in the API of proprietary LLMs like OpenAI's GPT series. These changes might pose difficulties for future reproducibility of our results.
>
> To mitigate these concerns, we will provide all the implementation details (see our response to Q4 as an example). We will specify the time stamps for our experimental results for each proprietary LLM in the revised paper. This will give clarity about the version and configuration of the models we used. By openly sharing our benchmark and codebase, we encourage the community to periodically evaluate LLMs. This would help in tracking the performance trajectory of models over time.
>
> > Q2: What would be human performance? The task is quite easy, but I'm not sure if the average human rater (not connected with academia) would have a 100% success rate.
>
> We randomly selected 300 samples from the dataset and employed three annotators (native English speakers) to complete the human evaluation.
>
> The results show that with a little hint (refer to prompt 5# and 6#), human participants achieved a success rate close to 100%. While this task is unchallenging for human, advanced LLMs have suffered obstacles. We believe that this observation can help guide the development of subsequent LLMs.
>
> > Q3: Why, in the benchmark, every relation needs so many entity pairs? The entities don't influence how the question/instruction/prompt is phrased. Did you observe that different entities/pairs of entities for the same relation lead to different results?
>
> Yes, we have observed that different entity pairs for the same relation lead to different results. We take relation “hypernym” in prompt 4# of Re2Text task as an example and present analysis. In the following, Table 1 shows the accuracy of different models in this setting, and Table 2 shows the overlap percentage of incorrect answers (entity pairs) between different models. The low overlap indicates that different entity pairs for the same relation indeed lead to different results on different models.
>
> **Table 1**
>
> | Model Name | gpt-3.5-turbo | gpt-4 | claude-1 | claude-instant-1 | flan-t5-xxl |
> | --- | --- | --- | --- | --- | --- |
> | Accuracy | 77.50% | 71.25% | 100% | 83.75% | 47.50% |
>
> **Table 2**
>
> | Overlap of incorrect entity pairs | gpt-3.5-turbo | gpt-4 | claude-1 | claude-instant-1 | flan-t5-xxl |
> | --- | --- | --- | --- | --- | --- |
> | gpt-3.5-turbo | - | 11.25% | 0.0% | 1.25% | 17.5% |
> | gpt-4 | 11.25% | - | 0.0% | 2.5% | 25% |
> | claude-1 | 0.0% | 0.0% | - | 0.0% | 0.0% |
> | claude-instant-1 | 1.25% | 2.5% | 0.0% | - | 0.0% |
> | flan-t5-xxl | 17.5% | 25% | 0.0% | 0.0% | - |
>
> We will add a more comprehensive analysis in the appendix of the revised paper.
>
> > Q4: What version of ChatGPT/GPT4 was used (from which date)?
>
> The version of ChatGPT we used is `gpt-3.5-turbo-0301`. In the implementation, the `model` parameter should be set to `gpt-3.5-turbo-0301` instead of `gpt-3.5-turbo` when calling the OpenAI API. And the version of GPT4 we used is `gpt-4-0314` .
>
> > Q5: Figure 6 legends is too small to read on paper.
>
> We will make the corresponding modifications in the revised paper.

---

### Official Review · Reviewer_RTYu · 2023-08-02

**Soundness:** 4

**Excitement:**

3: Ambivalent: It has merits (e.g., it reports state-of-the-art results, the idea is nice), but there are key weaknesses (e.g., it describes incremental work), and it can significantly benefit from another round of revision. However, I won't object to accepting it if my co-reviewers champion it.

**Paper Topic And Main Contributions:**

In this work, the authors conduct a thorough investigation into the ability of Large Language Models (LLMs) to comprehend converse relations. Converse relations are defined as the opposite of a given semantic relation. For example, in the normal relation "x has part y," its converse form would be "y has part x." The authors introduce a novel benchmark, called ConvRe, to evaluate LLMs' capabilities in correctly matching relations with their corresponding natural language text in both normal and converse forms.
The study focuses on three LLMs: OpenAI's GPT, Anthropic's Claude, and Google's Flan-T5. Through a series of experiments, the authors assess the models' performance under various settings, including zero-shot, few-shot, relations-to-text, text-to-relations, and different model sizes. They analyze the obtained results to gain insights into the reasoning abilities of the examined LLMs.

**Reasons To Accept:**

- The paper presents an interesting analysis of the reasoning capabilities of LLMs by measuring their ability to understand converse relations.
- The authors introduce a new benchmark (ConvRe) for normal and converse relations to assess the semantic comprehension capabilities of future LLMs.

**Reasons To Reject:**

- While this work presents an interesting analysis of the reasoning capabilities of LLMs (even if a few specific) and proposes a benchmark for assessing future language models, I believe that the paper's content might be diluted for its length, and condensing it into a shorter paper could potentially better convey its findings. There appears to be some redundancy among Figures 2, 3, and 4, as they share a lot of information within themselves. Additionally, the bottom part of Figure 1 seems somewhat uninformative, with only one line in the leftmost graph and models not clearly represented. For Figures 5 and 6, I would rather explore the possibility of using tables instead of presenting a whole page of graphs. I feel it might be easier for readers to follow and compare information.
- I feel like drawing a clear and concise conclusion from this work is challenging due to the vastly different behaviors exhibited by the three models under study. For instance, Claude shows, in general, insensitivity to the number of parameters, performing well or not regardless of this factor, while Flan-T5 aligns more closely with the initial hypothesis proposed by the authors.  Additionally, in the few-shot setting, Claude models display completely opposite behavior in the Re2Text and Text2Re tasks, whereas GPT and Flan-T5 models exhibit more consistency. Why is that so? Is it because of the data used in their pre-training? Is it because of a difference in the number of parameters of these two different models (how claude-1 compares with Flan-T5 XXL?)?

**Reproducibility:**

3: Could reproduce the results with some difficulty. The settings of parameters are underspecified or subjectively determined; the training/evaluation data are not widely available.

**Reviewer Confidence:**

4: Quite sure. I tried to check the important points carefully. It's unlikely, though conceivable, that I missed something that should affect my ratings.

---

> ### Author Rebuttal · Authors · 2023-08-29
>
> Thank you for your detailed comments, and they are really helpful for us to improve our paper. We will carefully incorporate them in the revised paper.
>
> > Q1: The paper's content might be diluted for its length. There appears to be some redundancy among Figures 2, 3, and 4.
>
> While the textual contents of Figure 2 and 3 may seem similar, they hugely differ in their underlying challenges. For Re2Text task, the challenge lies in choosing the correct answers that are presented in less conventional manners (i.e., **altered text**). However, for Text2Re task, the challenge is to avoid being misled by the superficial patterns (i.e. **normal text**) in questions. So we presented the subtle but import details in two separate figures.
>
> Figure 4 is utilized to give a clear presentation of the prompt used in our experiments, including the hint. We will move this figure to appendix and make the main body more compact.
>
> > Q2: The bottom part of Figure 1 seems somewhat uninformative.
>
> Regarding the bottom part of Figure 1, we would like to highlight the unique challenges converse relations present for LLMs, potentially leading to diverse scaling trends. Concretely, LLMs exhibit positive scaling trend for normal relations. But for converse relations, LLMs show diverse scaling trends given different prompting methods. Yes, we will remove this part for the sake of clarity and brevity. This adjustment would also enable us to reintegrate some material from the appendix into the main text.
>
> > Q3: For Figures 5 and 6, I would rather explore the possibility of using tables instead of presenting a whole page of graphs.
>
> Figures 5 and 6 were introduced with the purpose of effectively illustrating the scaling trends of each model family under diverse settings. For example, the scaling pattern in Figure 6(f) is readily discernible. On the other hand, given the number of experimental settings we conducted (12 prompt settings per model across a total of 13 models for each task),  the table would be too big with too many numbers in it, which may make the table not quite reading friendly. We will provide the table of our entire experimental results in appendix for better clarity.
>
> Although both the 2Eqb reviewer and the moP5 reviewer mentioned that this paper is well organized and easy to follow, we will seriously consider your suggestions and do our best to make the paper better in revision.
>
> > Q4: I feel like drawing a clear and concise conclusion from this work is challenging due to the vastly different behaviors exhibited by the three models under study. For instance, Claude shows, in general, insensitivity to the number of parameters, performing well or not regardless of this factor, while Flan-T5 aligns more closely with the initial hypothesis proposed by the authors. Additionally, in the few-shot setting, Claude models display completely opposite behavior in the Re2Text and Text2Re tasks, whereas GPT and Flan-T5 models exhibit more consistency. Why is that so? Is it because of the data used in their pre-training? Is it because of a difference in the number of parameters of these two different models (how claude-1 compares with Flan-T5 XXL?)?
>
> While it is indeed challenging to draw concise conclusions about the scaling behaviors of the three models in question—Claude, Flan-T5, and GPT—our primary aim in this study is to demonstrate that these models do not adhere to a simple 'the bigger, the better' scaling law. The inconsistencies you pointed out, such as Claude's insensitivity to parameter size and its divergent behavior in different tasks, serve as supporting evidence for our main argument: that the current LMs lack a consistent and robust reasoning capability, particularly when assessed with converse relation understanding.
>
> Although investigating the effects of pretraining data and objectives on these behaviors is an important area for future research, those questions are beyond the scope of this particular study. Our focus is to challenge the prevailing belief that the success of LMs can be solely attributed to their generalized reasoning capacities.
>
> Finally, it's worth noting that arriving at a clear and comprehensive understanding of LMs is complicated by various factors. These include the unknown data sources and pretraining methods used for proprietary models (e.g., Claude and GPT) and limited availability of compute resources.
>
> We will add the discussion as well as the limitation in the revision for better clarity.

---

### Official Review · Reviewer_moP5 · 2023-08-03

**Soundness:** 5

**Excitement:**

4: Strong: This paper deepens the understanding of some phenomenon or lowers the barriers to an existing research direction.

**Missing References:**

Spurious correlation (in LM) might worth mentioning. Some papers to begin with:
Lifu Tu, Garima Lalwani, Spandana Gella, and He He. 2020. An Empirical Study on Robustness to Spurious Correlations using Pre-trained Language Models. Transactions of the Association for Computational Linguistics, 8:621–633.

Emily M. Bender, Timnit Gebru, Angelina McMillan-Major, and Shmargaret Shmitchell. 2021. On the Dangers of Stochastic Parrots: Can Language Models Be Too Big? In Proceedings of the 2021 ACM Conference on Fairness, Accountability, and Transparency (FAccT '21). Association for Computing Machinery, New York, NY, USA, 610–623. https://doi.org/10.1145/3442188.3445922

A side note: the authors might be interested in (and might have already been aware of) a new paper covering similar topic and testing more tasks, although it is **not** the authors’ obligation to cover or compare to concurrent papers.
Wu, Z., Qiu, L., Ross, A., Akyürek, E., Chen, B., Wang, B., Kim, N., Andreas, J., & Kim, Y. (2023). Reasoning or Reciting? Exploring the Capabilities and Limitations of Language Models Through Counterfactual Tasks. ArXiv, abs/2307.02477.

**Paper Topic And Main Contributions:**

This paper studies how good LLMs handle converse relations that are less common during training, and possible reasons and implications behind the performance drop. The main task is to test LLM’s ability to convert between text description and triples defined in formal language, while inverting the conventional meaning of the relations. Further tests on few-shot learning and “hard” paraphrases support that LLM rely on superficial correlations. Experiments conducted on LLMs of different scales confirm that the problem gets worse with larger LLMs.

**Questions For The Authors:**

Question A: Are “shortcuts”, “superficial correlations”, “superficial cues” the same thing as “spurious correlation”? If not what are their relations?

Question B: Just a random thought, not critical to the review - in the case of formal language, will it solve the problem if we change all symbols to more neutral names? For example, we rename “has part” to “relation A”.

**Reasons To Accept:**

The paper reveals an important shortcoming of large language model: the reliance on superficial correlations when dealing with some formal-language oriented tasks. What’s more important is that this problem doesn’t get solved (but gets worse) with scale.

Experiments are carefully-designed and provide sound and convincing justifications to the claims. The paper is well-written and easy to follow.

**Reasons To Reject:**

The scope of the study is relatively narrow - spurious correlation in LLM might be ubiquitous, and this paper only focuses on the case of relationship triple ↔ text conversion.

Minor:  the paper could be better contextualized among “shortcuts”, “superficial correlations”, “superficial cues”, “spurious correlation”, "counterfactual"

**Reproducibility:**

5: Could easily reproduce the results.

**Reviewer Confidence:**

4: Quite sure. I tried to check the important points carefully. It's unlikely, though conceivable, that I missed something that should affect my ratings.

**Typos Grammar Style And Presentation Improvements:**

Line 117-119: not a grammatical sentence

Line 135: what is a localized scope? does that mean entities of the same type are used in a single triple? (what’s described in section 2.5)

Line 186: Figure 2 **and** 3

Line 375, 378:

from ACLPUB paper formatting guidelines:
> Refrain from using full citations as sentence constituents. Instead of “(Gusfield, 1997) showed that … In (Gusfield, 1997), …” write “Gusfield (1997) showed that … In Gusfield (1997), …”

Table 5: row “parent of” has a layout mistake?

---

> ### Author Rebuttal · Authors · 2023-08-29
>
> Thank you for your comments and constructive suggestions. We are grateful that you are interested in our paper. We will provide comprehensive responses to your comments and questions.
>
> > Q1: The scope of the study is relatively narrow - spurious correlation in LLM might be ubiquitous, and this paper only focuses on the case of relationship triple ↔ text conversion.
>
> As we mentioned in our response to a related concern raised by Reviewer qoc2, our choice to focus on the relationship between triple ↔ text conversion, specifically converse relations, stems from the critical role such relations play in interpreting structural knowledge. Such understanding is pivotal in key downstream applications, especially when interfacing with knowledge graphs. The nuances of these relationships are essential for tasks such as question answering.
>
> Our study is a step towards shedding light on how LLMs deal with such structured information, especially in the presence of potential shortcuts (or spurious correlation, please see our response to Q2). The incorporation of a counterfactual setup in our experiments further differentiates our approach, aiming to provide insights into the task-level generalization abilities of LLMs in real-world scenarios.
>
> We do acknowledge that there's a broader horizon to this issue, beyond just the conversion between relationships and text. We're motivated to expand the boundaries of our investigation in future research, hoping to provide a more holistic understanding of how LLMs handle spurious correlations across varied tasks.
>
> > Q2: The phrasing of the phenomenon is unconventional. Are “shortcuts”, “superficial correlations”, “superficial cues” the same thing as “spurious correlation”? If not what are their relations? Missing References.
> >
>
> Thank you for your insightful question regarding the terminologies used in our paper. Here's a clarification that we found in corresponding papers.
>
> **Spurious correlation:** refers to the prediction rules that work for the majority examples but do not hold in general [1].
>
> **Superficial cues / biases / artifacts:** refer to unintended correlations between input and output in existing datasets, which are often introduced during data collection or human annotation [2,3,4].
>
> **Shortcuts:** refer to decision rules that perform well on standard benchmarks but fail to transfer to more challenging testing conditions, such as real-world scenarios [5].
>
> While these terms may have nuanced differences, their essence converges to the idea that models might exploit unintended patterns in datasets, particularly those evident in the majority of examples. This can harm their ability to generalize in open-world scenarios. Many papers in the field have proposed methods to identify and rectify this phenomenon, emphasizing the creation of real-world test sets. In alignment with this, our paper introduces textual variance in our benchmark, thereby serving as adversarial test sets.
>
> Moreover, we have incorporated the counterfactual assumption to assess the real task-level generalization capabilities of LLMs. We're also aware of the paper “Reasoning or Reciting? Exploring the Capabilities and Limitations of Language Models Through Counterfactual Tasks”. We are happy to see applications of counterfactual tests on LLMs in a broader context.
>
> We will definitely refine our phrasing to provide clarity around these terminologies in the revision and ensure that relevant papers will be appropriately cited.
>
> > Q3: Will it solve the problem if we change all symbols to more neutral names? For example, we rename “has part” to “relation A”.
>
> We would like to highlight that we indeed explored analogous setup during our experiments. For instance, we considered renaming the relation label "have part" as a symbolic representation, i.e., denoted by "@" in the text. Interestingly, all of these alternative approaches show similar results with the ones in our paper.
>
> To provide further insights, we present the outcomes of prompt 1#, 2#, 3#, 4#, 7# and 8# wherein we substituted all instances of relation text with "relation R". We use "relation R" instead of "relation A" for the concern that "relation A" may have a chance to bias the model towards favoring answer A. Due to the time limit, all the experiments are conduct on a subset (see Appendix C for more details) of our benchmark. The figure in the parentheses is the difference between the neutral relation naming and normal naming under the same setup and on the same subset.
>
> Note: prompt 8# does not contain results of Flan-T5, as it struggles to adhere to Chain-of-Thought instructions.
>
> **Table 1 Re2Text**
>
> |  | Prompt 1# | Prompt 2# | Prompt 3# | Prompt 4# | Prompt 7# | Prompt 8# |
> | --- | --- | --- | --- | --- | --- | --- |
> | text-ada-001 | 0.494 (-0.04) | 0.509 (-0.003) | 0.509 (-0.003) | 0.515 (+0.015) | 0.494 (-0.012) | 0.5 (+0.003) |
> | text-babbage-001 | 0.518 (+0.051) | 0.537 (+0.074) | 0.537 (-0.015) | 0.527 (+0.012) | 0.500 (-0.009) | 0.466 (-0.019) |
> | text-curie-001 | 0.527 (+0.006) | 0.463 (-0.031) | 0.448 (-0.046) | 0.439 (-0.037) | 0.500 (-0.009) | 0.500 (-0.009) |
> | text-davinci-003 | 0.857 (-0.006) | 0.567 (-0.134) | 0.659 (+0.074) | 0.259 (+0.027) | 0.101 (-0.054) | 0.774 (+0.094) |
> | gpt-3.5-turbo | 0.765 (-0.073) | 0.616 (+0.134) | 0.384 (-0.211) | 0.229 (+0.083) | 0.439 (+0.110) | 0.716 (+0.253) |
> | gpt-4 | 0.985 (-0.003) | 0.918 (-0.027) | 0.561 (+0.335) | 0.439 (+0.268) | 0.317 (+0.088) | 0.784 (-0.146) |
> | Claude-1 | 0.905 (+0.003) | 0.777 (-0.031) | 0.732 (+0.198) | 0.537 (+0.165) | 0.335 (-0.055) | 0.848 (+0.031) |
> | Claude-instant-1 | 0.762 (+0.073) | 0.613 (-0.152) | 0.485 (+0.113) | 0.384 (-0.122) | 0.558 (-0.104) | 0.777 (-0.144) |
> | FLAN-T5-Small | 0.546 (+0.025) | 0.488 (+0.015) | 0.546 (+0.006) | 0.494 (+0.046) | 0.506 (+0.027) | - |
> | FLAN-T5-Base | 0.796 (-0.042) | 0.329 (+0.061) | 0.665 (-0.039) | 0.201 (+0.049) | 0.488 (+0.049) | - |
> | FLAN-T5-Large | 0.634 (-0.061) | 0.430 (-0.012) | 0.558 (+0.003) | 0.378 (+0.094) | 0.253 (+0.146) | - |
> | FLAN-T5-XL | 0.875 (-0.046) | 0.546 (-0.201) | 0.518 (+0.216) | 0.210 (+0.131) | 0.183 (+0.137) | - |
> | FLAN-T5-XXL | 0.738 (-0.070) | 0.591 (-0.095) | 0.476 (+0.104) | 0.290 (+0.064) | 0.180 (+0.034) | - |
>
> **Table 2 Text2Re**
>
> |  | Prompt 1# | Prompt 2# | Prompt 3# | Prompt 4# | Prompt 7# | Prompt 8# |
> | --- | --- | --- | --- | --- | --- | --- |
> | text-ada-001 | 0.482 (-0.021) | 0.482 (-0.006) | 0.485 (+0.003) | 0.470 (-0.030) | 0.500 (-0.009) | 0.518 (-0.006) |
> | text-babbage-001 | 0.530 (+0.006) | 0.552 (+0.028) | 0.451 (-0.049) | 0.473 (-0.036) | 0.500 (-0.009) | 0.500 (-0.009) |
> | text-curie-001 | 0.488 (-0.024) | 0.500 (+0.021) | 0.543 (+0.004) | 0.509 (-0.055) | 0.500 (-0.009) | 0.500 (-0.012) |
> | text-davinci-003 | 0.695 (-0.140) | 0.640 (-0.144) | 0.591 (+0.232) | 0.564 (+0.174) | 0.756 (+0.268) | 0.921 (+0.171) |
> | gpt-3.5-turbo | 0.506 (-0.095) | 0.503 (+0.033) | 0.503 (+0.101) | 0.506 (0) | 0.418 (+0.025) | 0.512 (-0.107) |
> | gpt-4 | 0.945 (+0.006) | 0.899 (-0.058) | 0.512 (+0.357) | 0.494 (+0.351) | 0.588 (+0.475) | 0.674 (+0.186) |
> | Claude-1 | 0.506 (-0.299) | 0.351 (-0.356) | 0.857 (+0.241) | 0.860 (+0.272) | 0.677 (+0.451) | 0.378 (+0.195) |
> | Claude-instant-1 | 0.780 (-0.095) | 0.716 (-0.010) | 0.808 (+0.564) | 0.646 (+0.274) | 0.790 (+0.622) | 0.671 (+0.320) |
> | FLAN-T5-Small | 0.503 (+0.012) | 0.500 (+0.006) | 0.512 (+0.033) | 0.509 (+0.015) | 0.512 (-0.015) | - |
> | FLAN-T5-Base | 0.494 (-0.018) | 0.491 (-0.012) | 0.506 (-0.003) | 0.512 (-0.009) | 0.500 (-0.046) | - |
> | FLAN-T5-Large | 0.668 (-0.116) | 0.665 (-0.024) | 0.314 (+0.082) | 0.363 (+0.025) | 0.421 (+0.101) | - |
> | FLAN-T5-XL | 0.689 (-0.259) | 0.530 (-0.250) | 0.616 (+0.488) | 0.628 (+0.378) | 0.363 (+0.244) | - |
> | FLAN-T5-XXL | 0.841 (-0.135) | 0.784 (-0.134) | 0.253 (+0.243) | 0.290 (+0.208) | 0.183 (+0.156) | - |
>
> Referring to the presented tables, it can be inferred that altering symbols to adopt more neutral names generally shows various effects on the models. The performance of most models in prompt 3# and 4# (the challenging setup) is still around 50% or even worse.  Therefore, this technique cannot solve this problem.
>
> We do observe considerable improvements for some models on some specific setups, especially on the Text2Re task. Interestingly, on the Text2Re task, Claude models obtain pretty high results and consistent improvements on converse relations (prompt  3# 4# 7# 8#), but they also suffer the performance drop on normal relations (prompt 1# 2#). It suggests that, at least for Claude models, more neutral relation names can help to alleviate the spurious correlation, which have opposite effects on normal and converse relations.
>
> We appreciate the reviewer’s inspiring suggestions. Comprehensive experimental results on this issue and analysis will be added in the revised paper.
>
> > Q4: **Line 135: what is a localized scope?**
>
> By “localized scope”, we refer to the fact that both the normal and converse relation definitions are specific to the current example (i.e., example-specific prompts). These definitions are not included in the meta prompt and will not influence other examples.
>
> > Q5: **Table 5: row “parent of” has a layout mistake?**
>
> No, actually, we initially want to express that "parent of" is the combination of relation "mother of person" from NELL-ONE dataset and relation "father" from WikiData5M dataset. We will change this layout to two rows with the relation name "parent of (mother)" and "parent of (father)" to make it more clear.
>
> > Q6: Typos and layout issues.
>
> Thank you for your attentive reading. We will correct them in the revised paper.
>
>
> [1] Lifu Tu, Garima Lalwani, Spandana Gella, and He He. 2020. An Empirical Study on Robustness to Spurious Correlations using Pre-trained Language Models. Transactions of the Association for Computational Linguistics, 8:621–633.
>
> [2] Emily M. Bender, Timnit Gebru, Angelina McMillan-Major, and Shmargaret Shmitchell. 2021. On the Dangers of Stochastic Parrots: Can Language Models Be Too Big? In Proceedings of the 2021 ACM Conference on Fairness, Accountability, and Transparency (FAccT '21). Association for Computing Machinery, New York, NY, USA, 610–623.
>
> [3] Le Bras, Ronan, Swabha Swayamdipta, Chandra Bhagavatula, Rowan Zellers, Matthew Peters, Ashish Sabharwal, and Yejin Choi. 2020. Adversarial filters of dataset biases. In International conference on machine learning, pp. 1078-1088. PMLR, 2020.
>
> [4] Timothy Niven and Hung-Yu Kao. 2019. Probing Neural Network Comprehension of Natural Language Arguments. In Proceedings of the 57th Annual Meeting of the Association for Computational Linguistics, 4658-4664.
>
> [5] Robert Geirhos, Jörn-Henrik Jacobsen, Claudio Michaelis, Richard Zemel, Wieland Brendel, Matthias Bethge, and Felix A Wichmann. 2020. Shortcut learning in deep neural networks. Nature Machine Intelligence, 2(11):665–673.

---

### Official Review · Reviewer_qoc2 · 2023-08-04

**Soundness:** 3

**Excitement:**

3: Ambivalent: It has merits (e.g., it reports state-of-the-art results, the idea is nice), but there are key weaknesses (e.g., it describes incremental work), and it can significantly benefit from another round of revision. However, I won't object to accepting it if my co-reviewers champion it.

**Paper Topic And Main Contributions:**

This paper investigates the understanding of structured semantics in formal languages by Large Language Models (LLMs) using a new benchmark called ConvRe, which focuses on converse binary relations.  The benchmark consists of 17 relations and 1240 triples extracted from popular knowledge graph completion datasets and features two tasks, Re2Text and Text2Re, to evaluate LLMs' ability to match relations with associated text.  The paper conducts experiments on three LLM families and finds that LLMs often rely on shortcut learning and still face challenges on the proposed benchmark, indicating that they may not fully comprehend the structured semantics of formal languages.

**Reasons To Accept:**

1. The paper introduces a novel benchmark called ConvRe, focusing on converse binary relations, which addresses the question of whether LLMs understand the structured semantics of formal languages beyond their pre-training data.  This benchmark provides a specialized evaluation to assess LLMs' ability to match relations with associated text.

2. The paper presents a comprehensive evaluation protocol that includes different prompting methods, variants of test text, and few-shot example text.  This robust evaluation sheds light on the strengths and weaknesses of LLMs when dealing with structured semantics in formal languages.

3. The paper's findings reveal that LLMs often resort to shortcut learning and face challenges on the proposed benchmark.  This insight highlights the limitations of current LLMs and can guide future research to improve their understanding of formal language semantics, benefiting the NLP community by driving advancements in model design and evaluation.

**Reasons To Reject:**

1. The problem of short-cut learning of language models is important and interesting, but converse relations seem to be a minority term. I believe this work would be more valuable if it could be studied from a more general perspective.

2. Some details are not clear. For example (1) In Figure 2 and 3, in the LLM Pre-training corpus, the text corresponding to the triple is “x has a part called y” instead of the sentence “y has a part called x” in the given instruction. Or some other sentence? The input context can influence the model's decisions just as much as the pre-trained corpus.



**Reproducibility:**

3: Could reproduce the results with some difficulty. The settings of parameters are underspecified or subjectively determined; the training/evaluation data are not widely available.

**Reviewer Confidence:**

3: Pretty sure, but there's a chance I missed something. Although I have a good feel for this area in general, I did not carefully check the paper's details, e.g., the math, experimental design, or novelty.

---

> ### Author Rebuttal · Authors · 2023-08-29
>
> Thank you for your detailed comments, and they are really helpful for us to improve our paper. We will carefully incorporate them in the revision.
>
> > Q1: The problem of short-cut learning of language models is important and interesting, but converse relations seem to be a minority term. I believe this work would be more valuable if it could be studied from a more general perspective.
>
> Understanding converse relations is critical for ensuring that LLMs robustly and accurately interpret the structural knowledge, especially that embedded within triple data. In downstream applications, particularly when interfacing with knowledge graphs represented as triples (h, r, t), this ability becomes essential, as it directly impacts key tasks such as question answering.
>
> Moreover, our focus is not limited to identifying the tendencies of LLMs towards shortcut learning (or spurious correlations, as elaborated in our response to reviewer moP5 Q2). We've moved a step further by incorporating a counterfactual setup in our experiments. This aims to assess the real-world task-level generalization abilities of LLMs.
>
> While we believe that converse relations provide an in-depth perspective, we concur that broadening the scope of our investigation will indeed enrich our study's value. We are committed to delving into this wider perspective in our subsequent research endeavors.
>
> > Q2: Some details are not clear. For example (1) In Figure 2 and 3, in the LLM Pre-training corpus, the text corresponding to the triple is “x has a part called y” instead of the sentence “y has a part called x” in the given instruction. Or some other sentence? The input context can influence the model's decisions just as much as the pre-trained corpus.
>
> If our understanding is correct, the question asks about the potential shortcuts existed in the instruction and identifies a presentation issue in Figure 2 and 3. In Figure 2 and 3, the "LLM Pre-Training corpus" should present instantiated triples and their corresponding text, rather than the abstract form. For example, it should read “Triple: (sword, has part, hilt), Text: The sword has a part called hilt; etc.” instead of the generic “Triple: (x, has part, y), Text: x has a part called y”. The instruction may have an implicit shortcut (or spurious correlation) with the pre-training corpus. Such correlation will be eliminated when converse relation is considered. This can cause models to misinterpret the input, as evidenced by our experimental results. We appreciate your feedback and will revise Figures 2 and 3 for clarity in the next version.
>
> As you commented, **the input context can influence the model's decisions just as much as the pre-trained corpus.** Ideally, a well trained LLM should accurately interpret the given instruction and properly leverage its pre-trained knowledge to make reasonable predictions. However, our findings indicate that the pre-trained corpus exerts more influence. Specifically, when the instruction (input context) deviates from patterns of the pre-training corpus, the model appears more challenged in accurately comprehending the input context.

---

### Meta-Review · Area_Chair_X9nT · 2023-09-17

**Recommendation:** 4

**Metareview:**

This paper introduces a benchmark for evaluating LLMs' ability to understand converse relations that are less common during training. More specifically it pairs text description and triples defined in formal language, while inverting the conventional meaning of the relations.
Few-shot learning experiments conducted on multiple LLMs ( GPT, Claude, and Flan-T5)  confirm that counter intuitively the problem gets worse with larger LLMs.

Strength:
The paper reveals an important issue with LLMs: the reliance on superficial correlations when dealing with some formal-language oriented tasks.

Weakness:
The scope of the study is limited to specific type of data-- relationship triple ↔ text conversion.

---

### Decision · Program_Chairs · 2023-10-07

**Decision:**

Accept-Main

**Comment:**

This paper introduces a benchmark for evaluating LLMs' ability to understand converse relations that are less common during training. More specifically it pairs text description and triples defined in formal language, while inverting the conventional meaning of the relations.
Few-shot learning experiments conducted on multiple LLMs ( GPT, Claude, and Flan-T5)  confirm that counter intuitively the problem gets worse with larger LLMs.

Strength:
The paper reveals an important issue with LLMs: the reliance on superficial correlations when dealing with some formal-language oriented tasks.

Weakness:
The scope of the study is limited to specific type of data-- relationship triple ↔ text conversion.